# LMS: Learnable Maximum Spike with Optimal Spike Representation for High-Performance and Efficient Spiking Neural Networks

## Abstract

Spiking Neural Networks (SNNs) have garnered increasing attention due to their brain-inspired mechanisms. By encoding information with sparse binary spikes, they replace multiplications with additions, substantially reducing energy consumption. However, the binary spike emission inherently leads to significant information loss. In this work, we propose a Learnable Maximum Spike (LMS) neuron, which emits integer values during training and dynamically learns the maximum membrane potential for each layer based on its own membrane potential distribution, thereby determining the maximum integer value the neuron can emit (referred to as spike maximum). Additionally, we introduce a decay balancing coefficient that allows the spike maximum to adapt to the gradient and change in membrane potential distribution between the early and late stages of training, thereby further enhancing the network's performance. Finally, to preserve spike-driven inference, we transform the binary representation problem of emitted values into an integer programming problem, yielding an optimal spike representation of integers that minimizes energy consumption. Extensive experiments have validated the effectiveness of the proposed LMS neuron, which consistently outperforms current state-of-the-art methods on static datasets (CIFAR10, CIFAR100, ImageNet) and a neuromorphic dataset (CIFAR10-DVS). Furthermore, LMS requires less inference memory (**-7.16**%), shorter inference time (**-12.09**%), and lower energy consumption (**-61.9**%).

## 1 Introduction

Artificial Neural Networks (ANNs) have achieved remarkable breakthroughs across diverse domains, including image recognition (He et al., 2016; Dosovitskiy et al., 2020), speech recognition (Gulati et al., 2020; Radford et al., 2023), and natural language processing (Vaswani et al., 2017; Achiam et al., 2023), consistently setting new state-of-the-art benchmarks (Chen et al., 2024; Zhou et al., 2019; Cao et al., 2022). However, these advances come with larger models and higher computational complexity, resulting in substantially greater energy consumption during both training and inference. As an alternative, Spiking Neural Networks (SNNs), inspired by the event-driven communication of biological neurons, have emerged as a promising low-power solution (Yao et al., 2023b; Xing et al., 2024; Guo et al., 2023b; Luo et al., 2024; Fang et al., 2023). Unlike traditional ANNs, SNNs transmit information using discrete binary spikes (0/1), effectively replacing energy-intensive multiply-accumulate operations (MACs) with sparse accumulate operations (ACs). This event-driven processing paradigm enables substantial energy savings on neuromorphic hardware, with previous studies demonstrating that SNNs can achieve energy efficiency improvements of several orders of magnitude compared to their ANN counterparts (Akopyan et al., 2015; Davies et al., 2018; Frenkel et al., 2023).

However, a major limitation of SNNs is the significant information loss from quantizing membrane potentials into binary spikes. This inherently restrictive activation mechanism curtails the model's expressive capacity, frequently resulting in inferior performance compared to conventional ANNs. Several studies have been conducted to mitigate this limitation (Guo et al., 2023a; 2022a; Luo et al., 2024; Yao et al., 2025). Notable among these is the I-LIF model (Luo et al., 2024), which generalizes spike activation from binary to non-negative integers, thereby alleviating quantization errors and

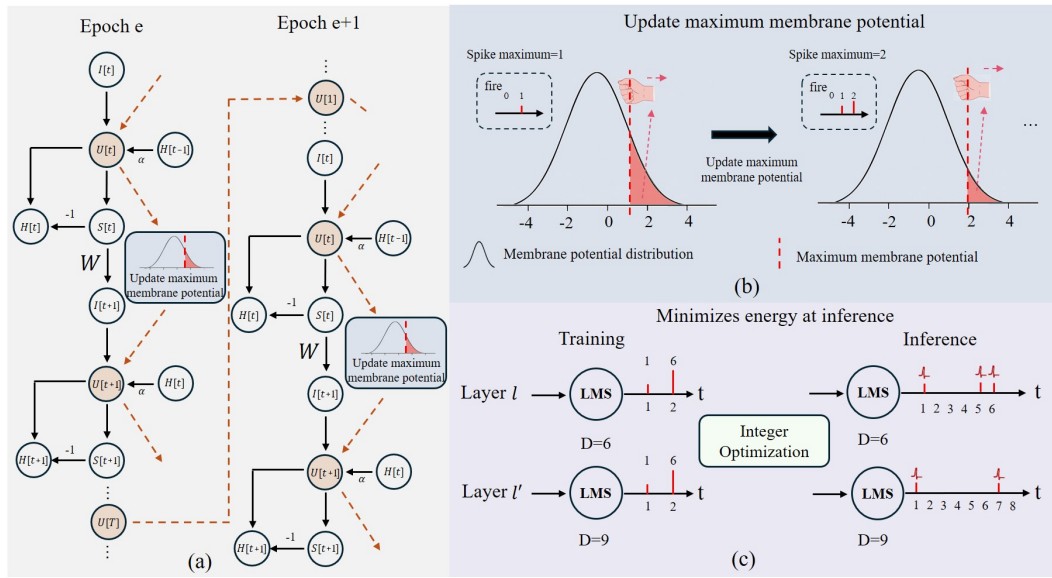

Figure 1: (a) illustrates the overall training pipeline of the proposed neuron, where the maximum membrane potential is updated dynamically. (b) details the update rule: whenever the membrane potential exceeds the current maximum, the excess is used to update the maximum, which in turn indirectly adjusts the spike maximum, making it better aligned with the membrane potential distribution of that layer. (c) presents the optimal spike representation. For each layer, given its spike maximum $D$, we formulate and solve an IP problem to derive the optimal spike representation that minimizes overall energy consumption.

strengthening the network's expressive capacity. Despite its effectiveness, the I-LIF model fails to consider the layer-wise variability in membrane potential distributions. Furthermore, a larger spike maximum (the maximum integer value a neuron can emit) leads to multiplicative increases in memory usage, inference latency, and energy consumption, posing practical challenges for deployment.

In this paper, we propose the Learnable Maximum Spike (LMS) neuron. As illustrated in Figure 1(a), the model enables each layer to adapt its maximum membrane potential dynamically during training. Furthermore, as depicted in Figure 1(b), each neuron updates this maximum membrane potential based on the current layer's membrane potential distribution, thereby adjusting the spike maximum to match the layer-specific membrane potential distribution better. Such layer-wise adaptation ensures that each layer acquires sufficient expressive capacity to capture its input dynamics, while avoiding unnecessary computational costs. Moreover, we introduce a decay balancing coefficient to dynamically regulate the update rate of the maximum membrane potential throughout the training process, enabling rapid adaptation to membrane potential distribution early on and stabilizing updates later to enhance overall performance. Finally, as depicted in Figure 1(c), we propose an optimal spike representation that converts neuronal firing integer values into binary (0/1) spikes during inference. By formulating the optimal spike representation of integer values in each layer as an integer programming (IP) problem, we derive an energy-minimizing spike configuration tailored to each layer.

Overall, the key contributions of our paper are summarized as follows:

- We propose a novel learnable maximum spike neuron that dynamically determines the layer-specific spike maximum based on membrane potential distribution, balancing the expressive capacity with computational and memory efficiency. Furthermore, we introduce a decay balancing coefficient that regulates the spike maximum update schedule across training to improve performance and convergence stability.

- To achieve efficient inference, we develop an optimal spike representation that minimizes energy consumption. By formulating spike representation as an IP problem, it maps integer firing values to binary spike sequences, effectively controlling energy usage even with increasing spike maximum and preventing energy consumption surges.

- Experimental results demonstrate state-of-the-art performance across both static and neuromorphic datasets. Our ResNet34-based model, for instance, achieves 71.36% top-1 accuracy on ImageNet, while outperforming other methods with fewer inference timesteps and significantly reduced energy consumption.

## 2 RELATED WORKS

Spiking neurons convert continuous membrane potentials into binary spikes, inevitably leading to substantial information loss. To alleviate this problem, Rmp-Loss (Guo et al., 2023a) introduces a loss function that encourages membrane potentials to approach both zero and the firing threshold, thereby mitigating the quantization error caused by binarization. Similarly, InfLoR-SNN (Guo et al., 2022b) employs a membrane potential rectifier that redistributes membrane potentials toward the firing threshold to reduce information degradation. Nevertheless, these approaches remain confined to binary spike emission, which fundamentally restricts neuronal expressiveness.

As an alternative to binary spikes, ternary spike (Sun et al., 2022; Guo et al., 2024a) extend the traditional binary format to either $\{0, 1, 2\}$ or $\{-\alpha, 0, \alpha\}$, providing an improvement in expressiveness. Nonetheless, the gains in expressiveness are marginal, and in the former case, the value 2 is not functionally distinguished, undermining SNNs' energy efficiency. More recently, the I-LIF model (Luo et al., 2024) generalizes spike firing values to non-negative integers at training and converts them into binary spikes at inference, significantly improving expressive capacity while preserving low-power ACs. However, it overlooks the heterogeneous membrane potential distributions across layers, and as the spike maximum increases, memory footprint and inference latency grow multiplicatively.

These limitations highlight open questions that remain underexplored: **(1) Should a fixed spike maximum be imposed uniformly across all layers in an SNN?** and **(2) Can layers with larger spike maximum be adaptively transformed into more compact yet energy efficient representations?**. Addressing these is crucial for balancing expressiveness and energy in practical SNNs deployment.

In summary, prior works have sought to mitigate information loss through refining binary spike generation or by extending spike values to ternary or integer schemes. Yet, these solutions either yield marginal (or no) expressiveness gains or incur substantial overhead. **In contrast, we revisit the design of both spike firing and representation, arguing that a uniform spike maximum across layers is unnecessary and that existing inference representations are suboptimal. To this end, we propose that each layer learn its own spike maximum based on its membrane potential distribution, and introduce a layer-wise integer programming framework to obtain an optimal spike representation. Together, these enhance the expressive capacity of spiking neurons while maintaining low computational costs.**

## 3 PRELIMINARY

**LIF.** Spiking neurons are the basic computational units of SNNs, inspired by biological neural processes. Among them, the Leaky Integrate-and-Fire (LIF) neuron (Nahmias et al., 2013), an improved variant of the Integrate-and-Fire (IF) neuron (Maass & Bishop, 2001), is the most widely used for its balance of biological plausibility and computational simplicity. The soft-reset LIF neuron is defined as:

$$U[t] = \tau H[t - 1] + I[t], \tag{1}$$
$$S[t] = \Theta(U[t] - V_{th}), \tag{2}$$
$$H[t] = U[t] - S[t], \tag{3}$$

where $U[t]$ is the membrane potential at timestep $t$, integrating temporal trace $H[t-1]$ and spatial input $I[t]$, $\tau$ is the decay factor, $\Theta(\cdot)$ is the Heaviside step function, and $V_{th}$ is the firing threshold. A spike is emitted if $U[t] > V_{th}$; otherwise, the neuron remains silent. After firing, $U[t]$ is reduced by the spike value, realizing a soft reset.

**I-LIF.** Since LIF generates only binary spikes (0/1), its representational capacity is limited. The Integer LIF (I-LIF) model (Luo et al., 2024) addresses this by extending outputs to non-negative integers, rewriting Equation (2) as:

$$S[t] = \text{clip}(\text{round}(U[t]), 0, D), \tag{4}$$

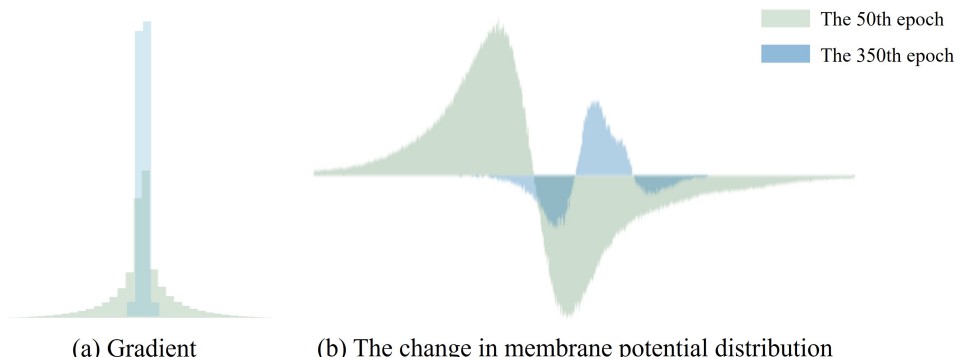

(a) Gradient        (b) The change in membrane potential distribution

Figure 2: Gradients and change in membrane potential distribution for ResNet20 on CIFAR100 trained for 400 epochs. (a) show the gradient of the 50th epoch and the 350th epoch. (b) show the membrane potential distribution shifts of the 50th epoch and the 350th epoch. Weight and membrane potential distributions shift much more rapidly in early training than in later stages.

where $\mathrm{clip}(x, 0, D)$ restricts $x$ to $[0, D]$, $\mathrm{round}(\cdot)$ is the rounding function, $D$ is a user-defined spike maximum. For $D = 1$, I-LIF reduces to LIF with a firing threshold of 0.5. During inference, integer $S^l[t]$ in layer $l$ are decomposed into $D$ binary spikes $\{S^l_d[t]\}^D_{d=1}$, with $S^l[t] = \sum^D_{d=1} S^l_d[t]$, extending inference from $T$ to $T \times D$ timesteps. By convolution linearity, the neuron's input at $l + 1$ layer is:

$$I^l[t] = \sum_{d=1}^{D} W^l S^l_d[t], \tag{5}$$

where $W^l$ is the weight of the $l$-th layer. This allows the use of efficient ACs instead of MACs.

A key limitation of I-LIF is its reliance on a fixed, layer-invariant $D$. Larger $D$ improves accuracy but increases latency and energy, while smaller $D$ reduces cost at the expense of performance. In practice, membrane potential distributions differ across layers (see Section A.1), making a uniform $D$ suboptimal: some layers are over-provisioned, wasting resources, while others are under-provisioned, reducing expressive capacity.

This observation reveals a critical open challenge: **should spike maximum be globally fixed, or tailored to each layer?** While the latter remains underexplored, it promises finer-grained accuracy–efficiency trade-offs. Motivated by this gap, we propose a natural yet powerful extension: allowing each layer to learn its own spike maximum $D$ dynamically. Such flexibility not only enhances representational capacity but also enables more adaptive and resource-efficient SNNs, paving the way toward high-performance and energy-efficient SNNs.

## 4 METHOD

Drawing on the above analysis, we introduce the **Learnable Maximum Spike LIF (LMS)** neuron, which enables each layer to learn its own spike maximum $D$. Once training is complete, a layer-wise IP problem is solved to obtain an energy-minimizing spike representation, thereby enhancing computational efficiency during inference. The overall algorithm is provided in Section A.2.

### 4.1 LEARNABLE MAXIMUM SPIKE

As illustrated in Figure 1(b), each neuron learns spike maximum $D$ by dynamically adjusting the maximum membrane potential. Specifically, we reformulate the firing mechanism in Equation (4) as:

$$S[t](M) = \mathrm{round}(\mathrm{clip}(U[t], 0, M)), \tag{6}$$

where $M$ is the learnable maximum membrane potential that need not be an integer, still fires an integer value. For each layer $l$, the maximum membrane potential is updated upon each firing according to:

$$M^l \leftarrow M^l + \alpha \frac{n^l}{N^l}, \tag{7}$$

where $n^l$ is the number of elements in layer $l$ whose membrane potentials exceed $M^l$, $N^l$ is the total number of elements in that layer, and $\alpha$ is a balancing coefficient that regulates the update rate of $M^l$.

The key idea behind this modification is to optimize the floating-point maximum $M$, which is more flexible and fine-grained, and then indirectly determines the spike maximum $D$ of layer $l$ via:

$$D^l = \text{round}(M^l). \tag{8}$$

When $M$ is an integer, Equation (4) and Equation (6) become functionally equivalent. By introducing $M$, each layer in the SNN can dynamically optimize its expressive capacity, achieving a balance between computational efficiency and expressive capacity.

Moreover, as illustrated in Figure 2, both the network weights and the membrane potential distributions undergo rapid changes in the early stages of training, whereas these changes slow down significantly in the later stages. Consequently, $M^l$ must update quickly at the beginning of training to promptly track the upper bound of the distribution, thereby preventing $D^l$ from becoming too small and causing information loss. Later in training, as weight and membrane potential updates stabilize, continuing to update $D^l$ at the same or higher rate may cause oscillations that hinder weight fine-tuning.

To this end, we use a balancing coefficient $\alpha$ that decays progressively over training. This enables fast updates of $M^l$ early on to track changing membrane potentials, and slower updates later to maintain stability. The decay is defined as:

$$\alpha_i = \frac{1}{2}\alpha_0(1 + cos(\frac{\pi i}{I})), \tag{9}$$

where $\alpha_i$ is the decay coefficient at epoch $i$, and $I$ is the total number of training epochs. We adopt the cosine decay strategy for its smooth and nonlinear descent, which maintains a high update rate during the early training stages and significantly reduces it in later stages, unlike the uniform reduction of linear decay, and our experiments confirm its superior effectiveness. This mechanism enables the network learns a sufficiently expressive spike maximum while maintaining convergence stability. From an information-theoretic perspective, we further show that the learnable maximum spike enhances the expressive capacity of the network while avoiding unnecessary consumption. The detailed analysis is provided in Section A.10.

### 4.2 OPTIMAL SPIKE REPRESENTATION

Using the LMS neuron model, we have effectively addressed a challenge. However, the spike maximum $D$ in certain layers may still become large. In such cases, continuing to use the I-LIF inference scheme would require excessive timesteps, making the inference process computationally expensive. This motivates us to explore whether the integer value can be transformed into an optimal spike representation during inference that preserves lossless performance while minimizing energy consumption and other computational costs.

For instance, when $D = 2^N - 1$, the standard binary representation is optimal. However, for arbitrary values of $D$, binary representation is generally suboptimal. To derive an optimal spike representation, we follow three key principles:

1. Employ weighted representation to ensure linear decodability after convolution;

2. Minimize the number of required timesteps;

3. Given a fixed number of timesteps, minimize the total number of "1"s across all binary representations to reduce energy consumption.

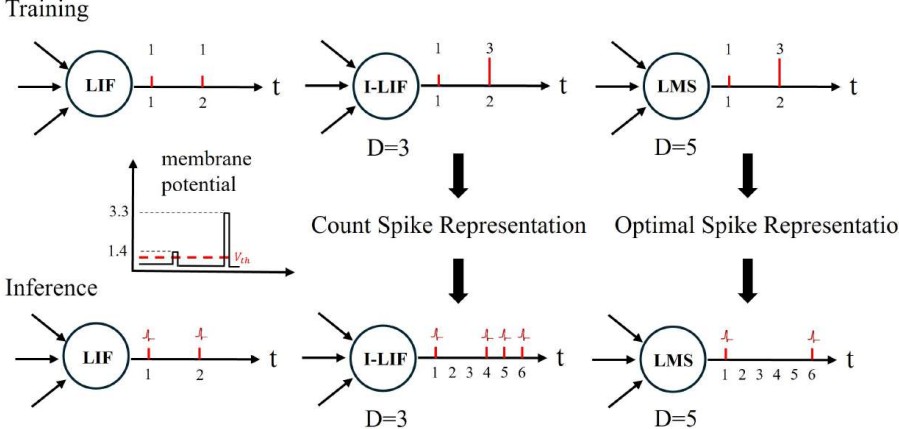

Figure 3: Comparison of LIF, I-LIF, and LMS. LIF emits binary spikes during both training and inference, introducing quantization errors. I-LIF and LMS emit integer values during training, and switch to count spike representation and optimal spike representation, respectively, during inference.

Based on these principles, the problem of finding the optimal spike representation for integers in the range $[0, D^l]$ within layer $l$ can be formulated as the following IP problem:

$$\min_{\mathbf{w}^l, \mathbf{B}^l} \quad \sum_{z=0}^{D^l} p_z^l \sum_{j=1}^{C^l} b_{j,z}^l \tag{10a}$$

$$\text{s.t.} \quad \sum_{j=1}^{C^l} w_j^l b_{j,z}^l = z, \quad \forall z = 0, 1, \dots, D^l, \tag{10b}$$

$$w_j^l \in \mathbb{Z}, \quad \forall j = 1, \dots, C^l, \tag{10c}$$

$$b_{j,z}^l \in \{0, 1\}. \tag{10d}$$

Here, $p_z^l = \frac{N_z^l}{N^l}$ denotes the probability of observing integer $z$ in the $l$-th layer of the training dataset, where $N_z^l$ is the number of neurons that emit integer $z$, and $N^l$ represents the total number of neurons emitting signals in the layer. The vector $\mathbf{w}^l = (w_1^l, w_2^l, \dots, w_{C^l}^l)$ contains the integer weights assigned to each inference timestep in layer $l$, with $w_j^l$ representing the weight for the $j$-th timestep. $C^l$ denotes the number of timesteps used for spike representation in layer $l$. To satisfy Principle2, we set $C^l = \lceil \log_2(D^l + 1) \rceil$. The codebook $\mathbf{B}^l \in \mathbb{R}^{C^l \times (D^l+1)}$ is a binary matrix where each element $b_{j,z}^l \in \{0, 1\}$ indicates whether a spike occurs at the $j$-th timestep when representing integer $z$ in layer $l$. The constraint $\sum_{j=1}^{C^l} w_j^l b_{j,z}^l = z$ ensures that the weighted sum of the per-timestep spike indicators $b_{j,z}^l$ by their corresponding weights $w_j^l$ satisfies the target integer value $z$.

By solving Problem (10), we obtain the optimal spike representation for each layer during inference. Specifically, this procedure converts $S^l[t](M)$ into a 0/1 spikes sequence $\{b_{j,k}^l[t]\}_{j=1}^{C^l}$. Based on this representation, the input of the neuron in the $(l+1)$-th layer can then be formulated as:

$$I^l[t] = W^l S^l[t](M) = W^l \sum_{j=1}^{C^l} w_j^l b_{j,z}^l[t] = \sum_{j=1}^{C^l} w_j^l (W^l b_{j,z}^l[t]), \tag{11}$$

Since $\{b_{j,z}^l[t]\}_{j=1}^{C^l}$ consists of binary (0/1) spikes, computing $W^l b_{j,z}^l[t]$ requires only ACs, thereby substantially reducing energy consumption. Figure 3 shows the differences among LIF, ILIF, and LMS. The solution method for the IP and detailed numerical results are provided in Section A.3, and we analyze the optimality of our method in Section A.10.

Table 1: Performance on CIFAR100. $T$ denotes the number of training timesteps. Since LMS produces different spike maximums across layers, $C$ indicates the average expansion timesteps of each training timestep during inference. For all other methods, $C$ is set to 1 unless otherwise specified.

| Method | Type | Architecture | T × C | Accuracy |
|---|---|---|---|---|
| RTS (Deng & Gu, 2021) | ANN2SNN | ResNet18 | 64 | 69.27% |
| SlipReLU (Jiang et al., 2023) | ANN2SNN | ResNet18 | 128 | 78.55% |
| Diet-SNN (Rathi & Roy, 2020) | SNN training | ResNet20 | 5 | 64.07% |
| Enof-SNN (Guo et al., 2024b) | SNN training | ResNet20 | 4 | 73.01% |
| ReverB-SNN (Guo et al., 2025) | SNN training | ResNet20 | 4 | 73.28% |
| Dspike (Li et al., 2021) | SNN training | ResNet20 | 4 | 73.35% |
| FSTA-SNN (Yu et al., 2025a) | SNN training | ResNet20 | 4 | 73.44% |
| TS-SNN (Yu et al., 2025b) | SNN training | ResNet20 | 4 | 73.46% |
| Ternary Spike (Guo et al., 2024a) | SNN training | ResNet20 | 4 | 74.02% |
| SLT-TET (Anumasa et al., 2024) | SNN training | ResNet19 | 6 | 74.87% |
| BKDSNN (Xu et al., 2024) | SNN training | ResNet19 | 4 | 74.95% |
| TEBN (Duan et al., 2022) | SNN training | ResNet19 | 6 | 76.41% |
| MI-TRQR (Xue et al., 2025) | SNN training | ResNet19 | 4 | 77.70% |
| InfLoR-SNN (Guo et al., 2022b) | SNN training | ResNet20 | 5 | 71.19% |
| | | ResNet19 | 4 | 78.42% |
| LMS (Ours) | SNN training | ResNet20 | 1 × 2.95 | **76.45%** ± 0.12 |
| | | | 2 × 3.16 | **76.62%** ± 0.05 |
| | | ResNet19 | 1 × 2.88 | **81.93%** ± 0.02 |
| | | | 2 × 3.17 | **82.11%** ± 0.10 |

## 5 EXPERIMENTS

We validate on three static datasets (CIFAR10 (Krizhevsky et al., 2009), CIFAR100 (Krizhevsky et al., 2009), and ImageNet (Deng et al., 2009)) as well as on a neuromorphic dataset (CIFAR10-DVS (Li et al., 2017)). The CIFAR10 dataset consists of $60,000$ images of size $32 \times 32$ pixels, divided into 10 classes, with $50,000$ images for training and $10,000$ for testing. CIFAR100 follows the same format but contains 100 classes. The ImageNet dataset is highly challenging, comprising approximately $1,300,000$ images of size $224 \times 224$ pixels, with $1,250,000$ images for training and $50,000$ for testing, across $1,000$ classes. CIFAR10-DVS is an event-driven dataset derived from CIFAR10, comprising $10,000$ event-frame images across 10 classes, with $9,000$ images for training and $1,000$ for testing. Top-1 accuracy is presented as the mean $\pm$ standard deviation over three runs. The learning method, additional experimental details, and the results of CIFAR10 are provided in Sections A.4 to A.6.

### 5.1 COMPARISON WITH STATE-OF-THE-ART METHODS

**CIFAR100.** As shown in Table 1, our method achieves accuracies of $76.45\%$ and $81.93\%$ on CIFAR100 with ResNet20 and ResNet19 architecture, respectively, requiring only 2.95 and 2.88 average inference timesteps. When using ResNet19, LMS outperforms SLT-TET (Anumasa et al., 2024) and BKDSNN (Xu et al., 2024) by $7.06\%$ and $6.98\%$. Compared to RTS Deng & Gu (2021) and SlipReLU (Jiang et al., 2023), two ANN2SNN approaches that require 64 and 128 timesteps to reach $69.27\%$ and $78.55\%$ accuracy, respectively, our method achieves $81.93\%$ with only 2.88 average inference timesteps, exceeding them by $12.66\%$ and $3.38\%$, respectively. Furthermore, when we increase the number of training timesteps, LMS continues to yield additional performance gains, demonstrating its scalability with respect to temporal depth.

**ImageNet.** Table 2 provides a detailed comparison of our method against existing methods on the ImageNet dataset. With only 3.00 and 3.33 average inference timesteps, our ResNet18 and ResNet34 models achieve $68.40\%$ and $71.36\%$ accuracy, respectively, establishing a clear performance advantage. For example, under the ResNet34 architecture, our method surpasses Real Spike Guo et al. (2022d) by $3.67\%$ ($71.36\% \ vs. \ 67.69\%$) and outperforms SEW ResNet Fang et al. (2021) by $4.32\%(71.36\% \ vs. \ 67.04\%)$. Importantly, our model is trained using only a single timestep, which

Table 2: Performance on the ImageNet dataset.

| Method | Type | Architecture | T × C | Accuracy |
|---|---|---|---|---|
| SRP (Hao et al., 2023a) | ANN2SNN | ResNet34 | 32 | 68.40% |
| MS-ResNet (Hu et al., 2024) | SNN training | ResNet18 | 6 | 63.10% |
| SML (Deng et al., 2023) | SNN training | ResNet18 | 4 | 64.53% |
| GAC-SNN (Qiu et al., 2024) | SNN training | ResNet18 | 6 | 65.14% |
| ReverB-SNN (Guo et al., 2025) | SNN training | ResNet18 | 4 | 66.58% |
| OTTT (Xiao et al., 2022) | SNN training | ResNet34 | 6 | 65.15% |
| InfLoR-SNN (Guo et al., 2022b) | SNN training | ResNet34 | 4 | 65.54% |
| SLTT (Meng et al., 2023) | SNN training | ResNet34 | 6 | 66.19% |
| RecDis-SNN (Guo et al., 2022c) | SNN training | ResNet34 | 6 | 67.33% |
| TKS (Dong et al., 2024) | SNN training | ResNet34 | 4 | 69.60% |
| FSTA-SNN (Yu et al., 2025a) | SNN training | ResNet34 | 4 | 70.23% |
| SEW ResNet (Fang et al., 2021) | SNN training | ResNet18 | 4 | 63.18% |
|  |  | ResNet34 | 4 | 67.04% |
| Real Spike (Guo et al., 2022d) | SNN training | ResNet18 | 4 | 63.68% |
|  |  | ResNet34 | 4 | 67.69% |
| LMS (Ours) | SNN training | ResNet18 | $1 \times 3.00$ | **68.40%** $\pm$ 0.08 |
|  |  | ResNet34 | $1 \times 3.33$ | **71.36%** $\pm$ 0.10 |

Table 3: Event-based classification results on the CIFAR10-DVS dataset.

| Method | Type | Architecture | T × C | Accuracy |
|---|---|---|---|---|
| Real Spike (Guo et al., 2022d) | SNN training | ResNet20 | 10 | 78.00% |
| ReverB-SNN (Guo et al., 2025) | SNN training | ResNet20 | 10 | 78.10% |
| CKA-SNN (Zhang et al., 2024) | SNN training | ResNet20 | 10 | 78.50% |
| STBP-tdBN (Zheng et al., 2021) | SNN training | ResNet19 | 10 | 67.80% |
| MLF (Feng et al., 2022) | SNN training | ResNet19 | 10 | 70.36% |
| DA-LIF (Zhang et al., 2025) | SNN training | ResNet19 | 10 | 78.00% |
| LM-H (Hao et al., 2023b) | SNN training | ResNet19 | 10 | 79.10% |
| Ternary Spike (Guo et al., 2024a) | SNN training | ResNet20 | 10 | 79.80% |
|  |  | ResNet19 | 10 | 79.80% |
| LMS (Ours) | SNN training | ResNet20 | $4 \times 2.79$ | **82.70%** $\pm$ 0.14 |
|  |  |  | $10 \times 3.10$ | **83.83%** $\pm$ 0.12 |

substantially reduces the overall training cost. These results confirm the effectiveness of our approach even on large-scale datasets.

**CIFAR10-DVS.** We further evaluate on the neuromorphic CIFAR10-DVS dataset, with results shown in Table 3. With a ResNet20 backbone, we achieve $82.70\%$ with $T = 4$ and $C = 2.79$. In comparison, Real Spike (Guo et al., 2022d) and CKA-SNN (Zhang et al., 2024) obtain $78.9\%$ and $78.5\%$ accuracy with 10 timesteps, meaning our method surpasses them by $3.8\%$ and $4.2\%$, respectively. Furthermore, LMS outperforms ResNet19-based STBP-tdBN (Zheng et al., 2021) and MLF (Feng et al., 2022) by $14.9\%$ and $12.34\%$, respectively, despite ResNet20 having a smaller model size. These results highlight the effectiveness and efficiency of our approach on neuromorphic datasets.

## 5.2 INFERENCE COST AND ENERGY CONSUMPTION EVALUATION

Energy consumption is a critical concern in SNNs. To evaluate the efficiency of our approach, we use CIFAR100 as a representative benchmark and compare the inference cost and energy consumption of LMS against LIF and I-LIF, using ResNet20 as the backbone. The corresponding results are summarized in Table 4, and the detailed methodology for energy calculation is provided in Section A.7. Although LMS introduces a few additional MAC operations during inference, as detailed in Equation (11), its overall energy cost remains minimal. Notably, LMS exhibits significant advantages across multiple dimensions: it reduces the average inference timesteps to 2.95, lowers memory usage

Table 4: Inference cost and energy consumption evaluation. $^*$ denotes the case without the optimal spike representation. In I-LIF, the $C = D$.

| Neuron | T × C | Inference memory | Inference time | Energy |
|---|---|---|---|---|
| LIF | $4 \times 1$ | 894M | 3.06s | 0.21mJ |
| I-LIF | $1 \times 4$ | 894M | 3.06s | 0.11mJ |
| LMS$^*$ | $1 \times 5.37$ | 1022M | 4.34s | 0.12mJ |
| **LMS** | $\mathbf{1 \times 2.95}$ | **830M** | **2.69s** | **0.08mJ** |

Table 5: Performance on the ImageNet dataset with transformer-based architecture.

| Method | Param | Energy | T × C | Accuracy |
|---|---|---|---|---|
| E-SpikeFormer (Yao et al., 2025) | 5.1M | 1.7mJ | $1 \times 4$ | 75.3% |
| E-SpikeFormer + LMS (Ours) | 5.1M | 1.3mJ | $1 \times 3.19$ | **76.6%** |

Table 6: Object detection results on the COCO dataset.

| Method | Param | Energy | T × C | mAP@50 | map@50:95 |
|---|---|---|---|---|---|
| SpikeYOLO (Luo et al., 2024) | 23.1M | 34.6mJ | $1 \times 4$ | 62.3% | 45.5% |
| SpikeYOLO + LMS (Ours) | 23.1M | 24.7mJ | $1 \times 3.15$ | **63.4%** | **46.7%** |

from $894M$ to $830M$, shortens inference time from $3.06s$ to $2.69s$, and cuts energy consumption to just $0.08$ mJ (only $38\%$ of LIF's and $72\%$ of I-LIF's). These results indicate that LMS achieves notable improvements in inference efficiency and energy savings while achieving higher accuracy (see Section 5.4), making it more efficient for SNNs. Moreover, we further compare LMS with and without the optimal spike representation. Without an optimal spike representation, LMS incurs significantly higher resource demands: memory usage increases to $1022\,MB$ ($+23\%$), inference time rises to $4.34s$ ($+61\%$), and energy consumption reaches $0.12$ mJ ($+50\%$). A more detailed layer-wise comparison of inference timesteps can be found in Section A.8. These results highlight the crucial role of optimal spike representation for achieving high efficiency.

## 5.3 SCALABILITY TO OTHER ARCHITECTURE AND TASKS

Our method can be extended to other architectures such as Transformer and to other tasks such as object detection. We validate this scalability by (1) integrating LMS into E-SpikeFormer (Yao et al., 2025) and (2) applying it to SpikeYOLO (Luo et al., 2024) for object detection.

**Scalability to transformer-based architecture.** We integrate LMS into E-SpikeFormer and evaluate it on the ImageNet dataset to assess its scalability to Transformer-based SNN architectures. As shown in Table 5, with 5.1M parameters, E-SpikeFormer+LMS achieves 76.6% accuracy with 3.19 average inference timesteps, outperforming the vanilla E-SpikeFormer (75.3% with 4 inference timesteps) by $+1.3\%$ accuracy. Moreover, our method requires only 1.3mJ energy consumption, achieving a 24% reduction compared to the original model. These results demonstrate that LMS can be seamlessly extended to Transformer-based SNN architectures, highlighting its broad applicability across various model architectures. Across these experiments, LMS consistently yields higher accuracy while requiring fewer inference timesteps and lower energy consumption.

**Scalability to object detection.** We evaluate LMS on the COCO dataset Lin et al. (2014) with SpikeYOLO to assess its scalability to object detection. As shown in Table 6, our method achieves 63.4% mAP@50 and 46.7% mAP@50:95 with 3.15 average inference timesteps, outperforming SpikeYOLO with 4 inference timsteps by $+1.1\%$ mAP@50 and $+1.2\%$ mAP@50:95. Moreover, our method requires only 24.7mJ energy consumption, compared to 34.6mJ for SpikeYOLO. These results show that our method can be effectively extended to object detection, demonstrating its potential applicability to other tasks.

Table 7: Ablation studies on neuron models and decay strategies for $\alpha$ (initial value 0.01; step decay halves $\alpha$ every 100 epochs).

| (a) Neuron models | | | | | (b) Decay strategies for $\alpha$ | | | |
|---|---|---|---|---|---|---|---|---|
| Dataset | Neuron | T × C | Acc | | Dataset | Decay | T × C | Acc |
| CIFAR100 | IF | 4 × 1 | 70.64% | | CIFAR100 | / | 1 × 3.15 | 75.58% |
| | LIF | 4 × 1 | 72.98% | | | step | 1 × 2.95 | 75.98% |
| | I-LIF | 1 × 4 | 74.73% | | | linear | 1 × 2.95 | 76.01% |
| | LMS | 1 × 2.95 | **76.43%** | | | cosine | 1 × 2.95 | **76.43%** |
| CIFAR10-DVS | IF | 10 × 1 | 79.40% | | CIFAR10-DVS | / | 4 × 2.89 | 81.70% |
| | LIF | 10 × 1 | 79.60% | | | step | 4 × 2.84 | 81.80% |
| | I-LIF | 4 × 4 | 80.50% | | | linear | 4 × 2.84 | 82.00% |
| | LMS | 4 × 2.79 | **82.70%** | | | cosine | 4 × 2.79 | **82.70%** |

## 5.4 ABLATION STUDY

**LMS effectiveness.** To assess the contribution of our proposed LMS, we conducted ablation studies on the CIFAR100 and CIFAR10-DVS datasets using a ResNet20 backbone. The results are presented in Table 7a. Because I-LIF and LMS require additional timesteps during inference, both I-LIF and LMS correspondingly use fewer training timesteps compared to IF and LIF. For CIFAR100, the baseline using the IF and LIF achieves an accuracy of 70.64% and 72.98%. Replacing the neuron with I-LIF achieves an accuracy of 74.73%. After applying LMS, accuracy increases to 76.43%, corresponding to gains of 5.79%, 3.45%, and 1.7% over the IF, LIF, and I-LIF models, respectively. For CIFAR10-DVS, substituting LMS for IF, LIF, and I-LIF yielded accuracy improvements of 3.3%, 3.1%, and 2.2%, respectively. These results show that while I-LIF's integerized firing offers clear benefits, LMS delivers additional and more stable improvements through its learnable layer-wise spike maximum. By dynamically aligning each layer's spike range with evolving membrane potential statistics, LMS consistently enhances performance under the same or lower training budget, with gains generalizing across both static and neuromorphic datasets.

**Analysis of the Decay Strategy for the Balance Coefficient.** In Table 7b, we present ablation studies on different decay strategies for the balance coefficient $\alpha$. We evaluate no decay, step decay, linear decay (to 0), and cosine annealing decay (to 0) on CIFAR100 and CIFAR10-DVS using ResNet20. The results consistently show that cosine annealing decay outperforms the others. Specifically, on CIFAR100, it improves accuracy by 0.85%, 0.45%, and 0.42% over no decay, step decay, and linear decay, respectively. On CIFAR10-DVS, the corresponding improvements are 1.0%, 0.9%, and 0.7%. Additionally, the cosine annealing decay requires the fewest inference timesteps among all strategies, and all decaying strategies outperform the no-decay strategy. These results highlight the importance of the decay balance coefficient, with cosine annealing decay achieving the best performance by enabling rapid updates of $M$ in early training to track fast membrane potential changes, and gradually reducing the update rate in later stages to prevent unnecessary oscillations.

## 6 CONCLUSION

In this paper, we investigate layer-wise variations in membrane potential distributions and introduce the Learnable Maximum Spike (LMS) neuron, which enables each layer to dynamically learn its own spike maximum. By analyzing gradient dynamics and membrane potential distributions across early and late training phases, we adopt a cosine annealing strategy to update the maximum membrane potential, enabling rapid adaptation to fast-changing distributions during early training while avoiding unnecessary oscillations in later stages. In addition, integer value (0–D) are converted into optimal binary spike representations via an integer programming formulation, preserving energy efficiency during inference. Extensive experiments show that LMS not only delivers significant accuracy improvements but also reduces inference memory, inference time, and energy consumption. We believe this work will further promote the adoption of integer-valued firing and provide insights for unlocking the potential of SNNs.

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

## A APPENDIX

### A.1 MEMBRANE POTENTIAL DISTRIBUTIONS ACROSS LAYERS

We analyze the layer-wise membrane potential distributions in a ResNet20 model trained on CIFAR100 under three activation functions: LIF ($T = 4$), I-LIF ($T = 1, D = 4$), and ReLU (where the membrane potentials correspond to its pre-activation values).

As shown in Figure 4, regardless of the activation function, there are significant differences in the membrane potential distributions across layers, implying that a single, fixed spike maximum for each layer is suboptimal. Consequently, we propose that each layer should dynamically learn its own spike maximum according to its specific membrane potential distribution.

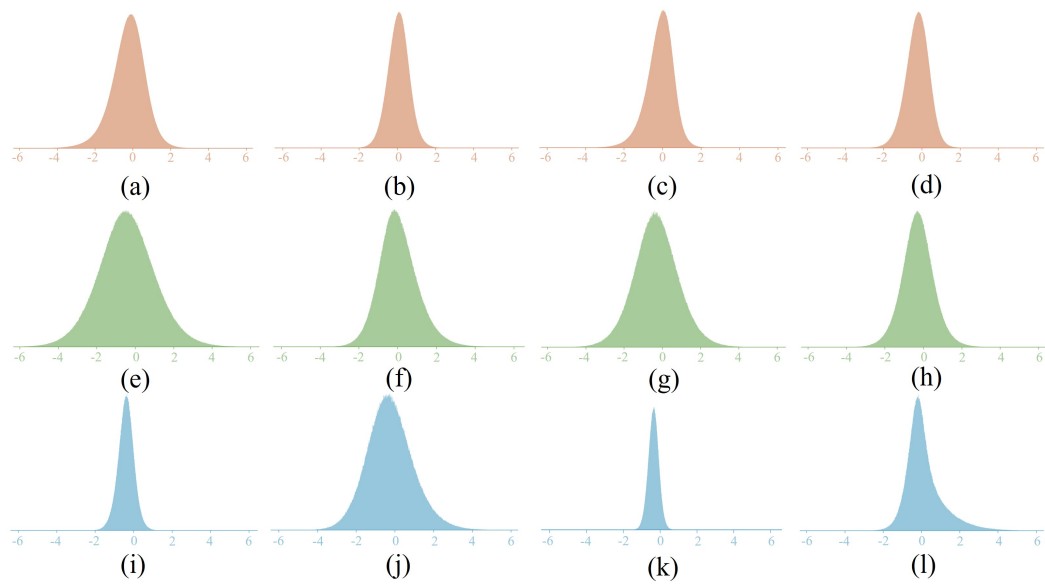

Figure 4: The membrane potential distributions of ResNet20 on CIFAR100. (a), (b), (c), and (d) are the distributions in the third stage using LIF neurons with 4 timesteps. (e), (f), (g), and (h) are the distributions in the third stage using I-LIF (D=4) neurons with 1 timesteps. (i), (j), (k), and (l) are the distributions in the third stage using ReLU.

### A.2  ALGORITHM

In our method, the decay balance coefficient $\alpha$ is updated once per epoch, and the maximum membrane potential $M$ is updated at each firing. The complete training procedure is shown in Algorithm 1.

---

**Algorithm 1** Training LMS

---

**Input:** Training dataset; total number of training iterations per epoch $E_{\text{train}}$; number of convolutional layers in the SNN $L$; maximum membrane potential for each layer $\{M^l\}_{l=1}^L$; initialize balance coefficient $\alpha_0$; total number of epochs $E$; training timestep $T$.
**Output:** The trained SNN ($D^l = round(M^l)$).
1: **for** $e = 1$ to $E$ **do**
2:     Update $\alpha_t = \frac{1}{2}\alpha_0(1 + \cos(\frac{\pi e}{E}))$;
3:     **for** $i = 1$ to $E_{\text{train}}$ **do**
4:         Obtain a mini-batch of training data and class labels;
5:         **for** $l = 1$ to $L$ **do**
6:           **for** $t = 1$ to $T$ **do**
7:             Perform forward propagation;
8:             Compute $n^l \leftarrow$ number of membrane potentials in layer $l$ exceeding $M^l$;
9:             Compute $N^l \leftarrow$ total number of elements in layer $l$;
10:            Update $M^l \leftarrow M^l + \alpha_t \frac{n^l}{N^l}$;
11:           **end for**
12:         **end for**
13:         Forward pass through the classification layer and compute loss;
14:         Backpropagate and update model parameters;
15:     **end for**
16: **end for**
17: Solving the IP problem (10) to obtain an optimal spike representation for each layer.

---

### A.3 Solution Method for Optimal Spike Representation and Numerical Results

#### A.3.1 Solving the Integer Programming (IP) Problem

The integer programming (IP) problem (10) can be reformulated as a mixed-integer linear programming problem and solved efficiently using the Branch-and-Cut algorithm (Padberg & Rinaldi, 1991). In particular, the nonlinear constraint $\sum_{j=1}^{C} w_j b_{j,k} = k$ can be equivalently replaced with the following set of linear constraints:

$$
\begin{aligned}
z_{j,k} &\leq w_j, && \forall j = 1, \ldots, C, \ \forall k = 0, \ldots, D, \\
z_{j,k} &\leq D\, b_{j,k}, && \forall j = 1, \ldots, C, \ \forall k = 0, \ldots, D, \\
z_{j,k} &\geq w_j - D\,(1 - b_{j,k}), && \forall j = 1, \ldots, C, \ \forall k = 0, \ldots, D, \\
z_{j,k} &\geq 0, && \forall j = 1, \ldots, C, \ \forall k = 0, \ldots, D, \\
\sum_{j=1}^{C} z_{j,k} &= k, && \forall k = 0, \ldots, D.
\end{aligned}
\tag{12}
$$

Note that each $w_j$ must satisfy $w_j \leq D$; otherwise, the formulation is invalid. In addition, as shown in Table 8, for CIFAR100 with ResNet20, the training stage requires approximately 17,580s, whereas the IP solver completes in 12.12s, accounting for only 0.069% of the total training time. For ImageNet with ResNet34, training takes about 326,080s, and the IP solve finishes in 19.01s, corresponding to merely 0.0058% of the overall training time. These results indicate that the IP solver overhead is negligible, thereby ensuring excellent computational efficiency. Looking ahead, we plan to develop specialized algorithms tailored to this IP problem, which may open up new directions for efficient discrete optimization in SNNs and further enhance scalability and performance.

Table 8: Total training time and IP solving time of our method under different datasets and models.

| Dataset | Model | GPU | Total training time | IP solving time |
|---|---|---|---|---|
| CIFAR100 | ResNet20 | Single RTX 4090 | 17580s | 12.12s |
| ImageNet | ResNet34 | Single RTX 4090 | 326080s | 19.01s |

#### A.3.2 Numerical Results

Our method generates optimal spike representations for ResNet20 on CIFAR100, as summarized in Table 9. The spike representation for each layer is determined using the layer-specific spike maximum $D^l$ and its corresponding firing rate. As shown in the results, earlier layers adopt larger spike maximums, while deeper layers employ smaller ones. This layer-wise variation underscores the importance of our approach, as it ensures every layer can fully leverage its expressive capacity without wasting computational resources.

### A.4 Learning Method for SNNs

Training strategies for high-performance SNNs can be broadly categorized into two paradigms. The first is ANN-to-SNN conversion method (ANN2SNN) (Bu et al., 2023; Rueckauer et al., 2017; Han et al., 2020; Hao et al., 2023a; Li & Zeng, 2022), which maps a pretrained ANN into an SNN by preserving weights while substituting activation functions with spiking neurons. Although effective in reusing mature ANN training pipelines, ANN2SNN typically requires many timesteps to approximate ANN performance (Jiang et al., 2023; Han & Roy, 2020) and is inherently incapable of surpassing its ANN counterpart. Moreover, it neglects the intrinsic temporal dynamics of spikes, thereby sacrificing one of the most distinctive features of SNNs.

The second paradigm is direct training, which circumvents the non-differentiability of spike generation through surrogate gradient methods. This approach allows effective optimization with fewer timesteps while leveraging temporal coding in SNNs (Liu et al., 2025; Hu et al., 2024; Guo et al., 2023c; Wu et al., 2018; Yao et al., 2023a). Owing to its efficiency and performance, direct training has become the dominant methodology and is also adopted in this work.

Table 9: The result of optimal spike representation on CIFAR100 using ResNet20

| Layer | D | C | Fire rating | Weight | Spike Representation |
|---|---|---|---|---|---|
| Layer1 | 12 | 4 | 0:0.5218; 1:0.3419; 2:0.0858; 3:0.0228; 4:0.0103; 5:0.0055; 6:0.0033; 7:0.0021; 8:0.0015; 9:0.0010; 10:0.0007; 11:0.0005; 12:0.0029 | (1,2,3,6) | 0:(0,0,0,0); 1:(1,0,0,0); 2:(0,1,0,0); 3:(0,0,1,0); 4:(1,0,1,0); 5:(0,1,1,0); 6:(0,0,0,1); 7:(1,0,0,1); 8:(0,1,0,1); 9:(0,0,1,1); 10:(1,0,1,1); 11:(0,1,1,1);12:(1,1,1,1) |
| Layer2 | 12 | 4 | 0:0.5928; 1:0.2795; 2:0.0814; 3:0.0267; 4:0.0092; 5:0.0041; 6:0.0022; 7:0.0015; 8:0.0008; 9:0.0004; 10:0.0003; 11:0.0002; 12:0.0010 | (1,2,3,6) | 0:(0,0,0,0); 1:(1,0,0,0); 2:(0,1,0,0); 3:(0,0,1,0); 4:(1,0,1,0); 5:(0,1,1,0); 6:(0,0,0,1); 7:(1,0,0,1); 8:(0,1,0,1); 9:(0,0,1,1); 10:(1,0,1,1); 11:(0,1,1,1); 12:(1,1,1,1) |
| Layer3 | 7 | 3 | 0:0.5628; 1:0.2709; 2:0.1087; 3:0.0390; 4:0.0133; 5:0.0040; 6:0.0010; 7:0.0003 | (1,2,4) | 0:(0,0,0); 1:(1,0,0); 2:(0,1,0); 3:(1,1,0); 4:(0,0,1); 5:(1,0,1); (0,1,1); (1,1,1) |
| Layer4 | 4 | 3 | 0:0.7092; 1:0.2453; 2:0.0393; 3:0.0054; 4:0.0008 | (1,2,3) | 0:(0,0,0); 1:(1,0,0); 2:(0,1,0); 4:(0,0,1); 5:(1,0,1) |
| Layer5 | 7 | 3 | 0:0.6441; 1:0.2075; 2:0.0918; 3:0.0384; 4:0.0130; 5:0.0038; 6:0.0010; 7:0.0003 | (1,2,4) | 0:(0,0,0); 1:(1,0,0); 2:(0,1,0); 3:(1,1,0); 4:(0,0,1); 5:(1,0,1); 6:(0,1,1); 7:(1,1,1) |
| Layer6 | 4 | 3 | 0:0.7955; 1:0.1635; 2:0.0335; 3:0.0063; 4:0.0012 | (1,2,3) | 0:(0,0,0); 1:(1,0,0); 2:(0,1,0); 4:(0,0,1); 5:(1,0,1) |
| Layer7 | 8 | 4 | 0:0.7126; 1:0.1411; 2:0.0756; 3:0.0390; 4:0.0184; 5:0.0080; 6:0.0033; 7:0.0013; 8:0.0008 | (1,2,3,4) | 0:(0,0,0,0); 1:(1,0,0,0); 2:(0,1,0,0); 3:(0,0,1,0); 4:(0,0,0,1); 5:(0,1,1,0); 6:(0,1,0,1); 7:(0,0,1,1); 8:(1,0,1,1) |
| Layer8 | 6 | 3 | 0:0.7056; 1:0.1967; 2:0.0675; 3:0.0208; 4:0.0063; 5:0.0020; 6:0.0012 | (1,2,3) | 0:(0,0,0); 1:(1,0,0); 2:(0,1,0); 3:(0,0,1); 4:(1,0,1); 5:(0,1,1); 6:(1,1,1) |
| Layer9 | 5 | 3 | 0:0.7762; 1:0.1665; 2:0.0452; 3:0.0097; 4:0.0019; 5:0.0004 | (1,2,3) | 0:(0,0,0); 1:(1,0,0); 2:(0,1,0); 3:(0,0,1); 4:(1,0,1); 5:(0,1,1) |
| Layer10 | 3 | 2 | 0:0.8453; 1:0.1328; 2:0.0192; 3:0.0026 | (1,2) | 0:(0,0); 1:(1,0); 2:(0,1); 3:(1,1) |
| Layer11 | 5 | 3 | 0:0.7729; 1:0.1509; 2:0.0544; 3:0.0160; 4:0.0042; 5:0.0015 | (1,2,3) | 0:(0,0,0); 1:(1,0,0); 2:(0,1,0); 3:(0,0,1); 4:(1,0,1); 5:(0,1,1) |
| Layer12 | 4 | 3 | 0:0.7508; 1:0.1929; 2:0.0451; 3:0.0091; 4:0.0021 | (1,2,3) | 0:(0,0,0); 1:(1,0,0); 2:(0,1,0); 3:(0,0,1); 4:(1,0,1) |
| Layer13 | 4 | 3 | 0:0.8036; 1:0.1541; 2:0.0346; 3:0.0064; 4:0.0013 | (1,2,3) | 0:(0,0,0); 1:(1,0,0); 2:(0,1,0); 3:(0,0,1); 4:(1,0,1) |
| Layer14 | 3 | 2 | 0:0.8769; 1:0.1107; 2:0.0112; 3:0.0011 | (1,2) | 0:(0,0); 1:(1,0); 2:(0,1); 3:(1,1) |
| Layer15 | 5 | 3 | 0:0.7858; 1:0.1558; 2:0.0447; 3:0.0106; 4:0.0024; 5:0.0007 | (1,2,3) | 0:(0,0,0); 1:(1,0,0); 2:(0,1,0); 3:(0,0,1); 4:(1,0,1); 5:(0,1,1) |
| Layer16 | 3 | 2 | 0:0.8100; 1:0.1609; 2:0.0256; 3:0.0035 | (1,2) | 0:(0,0); 1:(1,0); 2:(0,1); 3:(1,1) |
| Layer17 | 4 | 3 | 0:0.7915; 1:0.1482; 2:0.0465; 3:0.0111; 4:0.0027 | (1,2,3) | 0:(0,0,0); 1:(1,0,0); 2:(0,1,0); 3:(0,0,1); 4:(1,0,1) |
| Layer18 | 2 | 2 | 0:0.9163; 1:0.0783; 2:0.0054 | (1,2) | 0:(0,0); 1:(1,0); 2:(0,1) |
| Layer19 | 4 | 3 | 0:0.7631; 1:0.1407; 2:0.0623; 3:0.0232; 4:0.0107 | (1,2,3) | 0:(0,0,0); 1:(1,0,0); 2:(0,1,0); 3:(0,0,1); 4:(1,0,1) |

Table 10: Network architectures for CIFAR and ImageNet. $NC$ represents the number of classes. MP denotes max pooling, and AP denotes average pooling. All MP layers have a stride of 2. For the convolutional layer of Conv1, the CIFAR models employ a stride of 1, whereas the ImageNet models use a stride of 2.

| | CIFAR | | ImageNet | |
| --- | --- | --- | --- | --- |
| **Name** | **ResNet20** | **ResNet19** | **ResNet18** | **ResNet34** |
| Conv1 | $[3 \times 3, 64] \times 3$ 
 $3 \times 3$ MP | $3 \times 3, 128$ | $7 \times 7, 64$ 
 $3 \times 3$ MP | $7 \times 7, 64$ 
 $3 \times 3$ MP |
| Stage1 | $\begin{bmatrix} 3 \times 3, 64 \\ 3 \times 3, 64 \end{bmatrix} \times 2$ | $\begin{bmatrix} 3 \times 3, 128 \\ 3 \times 3, 128 \end{bmatrix} \times 3$ | $\begin{bmatrix} 3 \times 3, 64 \\ 3 \times 3, 64 \end{bmatrix} \times 2$ | $\begin{bmatrix} 3 \times 3, 64 \\ 3 \times 3, 64 \end{bmatrix} \times 3$ |
| Stage2 | $\begin{bmatrix} 3 \times 3, 128 \\ 3 \times 3, 128 \end{bmatrix} \times 2$ | $\begin{bmatrix} 3 \times 3, 256 \\ 3 \times 3, 256 \end{bmatrix} \times 3$ | $\begin{bmatrix} 3 \times 3, 128 \\ 3 \times 3, 128 \end{bmatrix} \times 2$ | $\begin{bmatrix} 3 \times 3, 128 \\ 3 \times 3, 128 \end{bmatrix} \times 4$ |
| Stage3 | $\begin{bmatrix} 3 \times 3, 256 \\ 3 \times 3, 256 \end{bmatrix} \times 2$ | $\begin{bmatrix} 3 \times 3, 512 \\ 3 \times 3, 512 \end{bmatrix} \times 2$ | $\begin{bmatrix} 3 \times 3, 256 \\ 3 \times 3, 256 \end{bmatrix} \times 2$ | $\begin{bmatrix} 3 \times 3, 256 \\ 3 \times 3, 256 \end{bmatrix} \times 6$ |
| Stage4 | $\begin{bmatrix} 3 \times 3, 512 \\ 3 \times 3, 512 \end{bmatrix} \times 2$ | / | $\begin{bmatrix} 3 \times 3, 512 \\ 3 \times 3, 512 \end{bmatrix} \times 2$ | $\begin{bmatrix} 3 \times 3, 512 \\ 3 \times 3, 512 \end{bmatrix} \times 3$ |
| | AP, NC-d FC | AP, NC-d FC | AP, 1000-d FC | AP, 1000-d FC |

## A.5 IMPLEMENTATION DETAILS

### A.5.1 NETWORK ARCHITECTURES

On CIFAR10 and CIFAR100, we adopt both ResNet19 (Zheng et al., 2021) and ResNet20 (Guo et al., 2022d) as backbone architectures. For CIFAR10-DVS, ResNet20 (Zheng et al., 2021) is also employed. On ImageNet, we use ResNet18 and ResNet34 (He et al., 2016). As summarized in Table 10, the initial feature extraction layer uses a 3×3 convolutional kernel for the CIFAR datasets, while a 7×7 kernel is utilized for ImageNet.

Moreover, during training, we take the average of the fully-connected layer's output over T timesteps as the network's final output, without requiring it to pass through an additional spiking neuron. This design allows the output to assume both positive and negative values. If we append another spiking neuron, whether in terms of the spike count or the average firing rate, the result could only be non-negative. Specifically, the network output is:

$$y_{out} = \frac{1}{T} \sum_{t=1}^{T} W x_t, \tag{13}$$

where $y_{out}$ denotes the output of network, $W$ is the weight matrix of the fully-connected layer, and $x_t$ is the network input at timestep $t$. During inference, it is sufficient to extend $T$ to $T \times C$, while keeping all other procedures unchanged.

### A.5.2 SURROGATE GRADIENT FUNCTION

Due to the non-differentiable nature of the spiking neuron's firing mechanism, backpropagation relies on a surrogate gradient. In this work, we employ the straight-through estimator (STE) (Rathi et al., 2020) as the surrogate gradient method, defined as follows:

$$\phi(U) = \begin{cases} 1, & if \quad 0 \leq U \leq M \\ 0, & otherwise \end{cases} \tag{14}$$

where $U$ is the membrane potential of the spiking neuron, and $M$ is the maximum membrane potential of the spiking neuron.

### A.5.3 EXPERIMENTAL SETTINGS

We train our models on CIFAR10, CIFAR100, and CIFAR10-DVS for 400 epochs using stochastic gradient descent (SGD) with a batch size of 64, an initial learning rate of 0.1, a momentum of 0.9, and a weight decay of 1e-4. The learning rate is decayed to zero via a cosine annealing

Table 11: Performance of image classification on CIFAR10 dataset.

| Method | Type | Architecture | T × C | Accuracy |
|---|---|---|---|---|
| SpikeNorm (Sengupta et al., 2019) | ANN2SNN | VGG16 | 2500 | 91.55% |
| SlipReLU (Jiang et al., 2023) | ANN2SNN | ResNet18 | 4 | 94.59% |
| KDSNN (Xu et al., 2023) | SNN training | ResNet18 | 4 | 93.41% |
| Real Spike (Guo et al., 2022d) | SNN training | ResNet20 | 5 | 93.01% |
| Dspike (Li et al., 2021) | SNN training | ResNet20 | 4 | 93.66% |
| CKA-SNN (Zhang et al., 2024) | SNN training | ResNet20 | 4 | 94.27% |
| FSTA-SNN (Yu et al., 2025a) | SNN training | ResNet20 | 4 | 94.72% |
| Ternary Spike (Guo et al., 2024a) | SNN training | ResNet20 | 4 | 94.96% |
| TEBN (Duan et al., 2022) | SNN training | ResNet19 | 6 | 94.71% |
| TAB (Jiang et al., 2024) | SNN training | ResNet19 | 6 | 94.81% |
| LSG (Lian et al., 2023) | SNN training | ResNet19 | 4 | 95.17% |
| SLT-TET (Anumasa et al., 2024) | SNN training | ResNet19 | 6 | 95.26% |
| RMP-Loss (Guo et al., 2023a) | SNN training | ResNet20 | 4 | 91.89% |
| | | ResNet19 | 4 | 95.51% |
| LMS (Ours) | SNN training | ResNet20 | $1 \times 2.73$ | $\mathbf{95.43\%} \pm 0.11$ |
| | | | $2 \times 3.05$ | $\mathbf{95.78\%} \pm 0.02$ |
| | | ResNet19 | $1 \times 2.64$ | $\mathbf{97.01\%} \pm 0.02$ |
| | | | $2 \times 2.88$ | $\mathbf{97.25\%} \pm 0.05$ |

schedule (Loshchilov & Hutter, 2017). For CIFAR10-DVS, input frames are resized to $48 \times 48$ pixels following (Guo et al., 2024a). On ImageNet, we train for 320 epochs with a batch size of 2048, while keeping all other optimizer settings consistent with those used for CIFAR100. For static datasets, input images are converted into spike sequences by the first layer of the network, whereas for neuromorphic datasets, raw spike data are fed directly into the model. All experiments are performed on a server equipped with a 64-core Intel Xeon Gold 6430 3.40 GHz CPU and 6 NVIDIA GeForce RTX 4090 GPUs.

### A.6 COMPARISON WITH SOTA METHODS ON CIFAR10 DATASET

On CIFAR10, as shown in Table 11, our method achieves $97.01\%$ accuracy with an average of 2.64 inference timesteps under the ResNet19 architecture, outperforming SLT-TET (Anumasa et al., 2024) and RMP-Loss (Guo et al., 2023a) by $1.75\%$ and $1.5\%$, respectively. Moreover, even when using the more efficient ResNet20 backbone, our approach still surpasses methods such as TEBN (Duan et al., 2022), TAB (Jiang et al., 2024), and LSG (Lian et al., 2023), all of which are based on ResNet19. It is worth noting that ResNet19 contains twice as many channels and consumes over $10 \times$ more energy than ResNet20 (Guo et al., 2023b). These results clearly demonstrate the effectiveness and efficiency of our proposed method.

### A.7 ENERGY CONSUMPTION CALCULATIONS

In conventional artificial neural networks (ANNs), each computation typically involves a multiply–accumulate (MAC) operation, whereas in spiking neural networks (SNNs), neurons emit binary spikes (0 or 1), and computations consist solely of accumulate (AC) operations. Based on 45 nm CMOS technology (Horowitz, 2014), a single MAC operation consumes 4.6 pJ, while an AC operation requires only 0.9 pJ, which corresponds to approximately one-fifth the energy of a MAC. The energy consumption for ANNs is calculated as follows:

$$E_{Conv}^{ANN} = F_{out} \times c_{in} \times c_{out} \times k^2 \times E_{MAC}, \tag{15}$$

$$E_{Linear}^{ANN} = f_{in} \times f_{out} \times E_{MAC}, \tag{16}$$

where $F_{out}$ is the spatial size of the output feature map, $c_{in}$ and $c_{out}$ are the numbers of input and output channels, $k$ is the kernel size, and $f_{in}$ and $f_{out}$ denote the input and output units of the fully connected layer, respectively. $E_{MAC} = 4.6$ pJ denotes the energy per MAC operation.

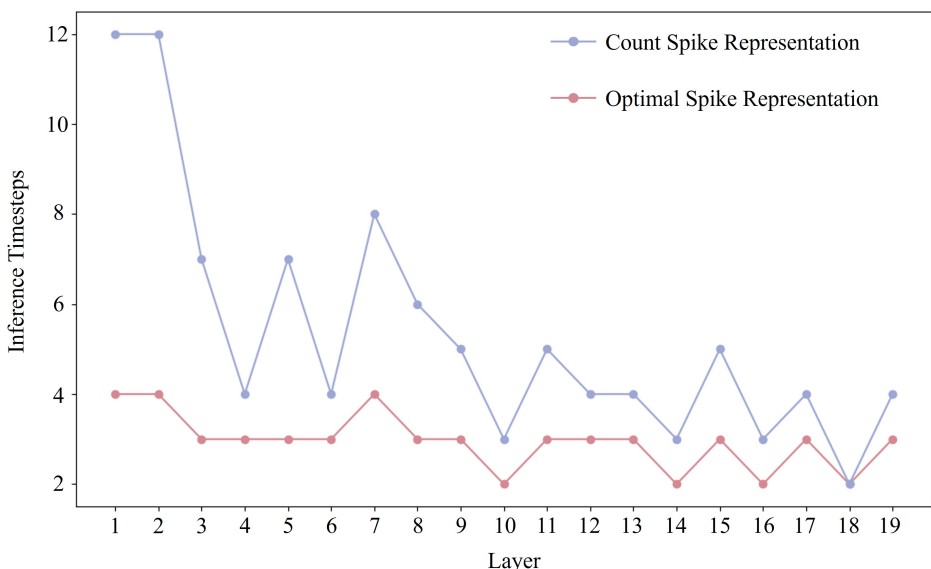

Figure 5: Inference timestep comparison of layer-by-layer for optimal spike representation and count spike representation.

For SNNs, the energy consumption is given by:

$$E_{Conv}^{SNN} = T \times C \times fr \times F_{out} \times c_{in} \times c_{out} \times k^2 \times E_{AC}, \tag{17}$$

$$E_{Linear}^{SNN} = T \times C \times fr \times f_{in} \times f_{out} \times E_{AC}, \tag{18}$$

where $fr$ denotes the average firing rate of a neuron over all timesteps, and $E_{AC} = 0.9$ pJ is the energy per AC operation. These equations highlight that the energy efficiency of SNNs stems from the use of low-energy AC operations and the sparsity of spike activations.

According to Eq. 11, our method introduces a small number of additional MAC and AC operations in the convolutional layers. Specifically, compared to standard SNNs, each convolution in our approach requires an extra $T \cdot C$ MAC operations and $T \cdot (C - 1)$ AC operations applied to the output feature map. The energy consumption of our method is thus expressed as:

$$E_{Conv}^{Ours} = (C \times fr \times F_{out} \times c_{in} \times c_{out} \times k^2 + (C - 1) \times F_{out} \times c_{out}) \times T \times E_{AC}$$
$$+ T \times C \times F_{out} \times c_{out} \times E_{MAC} \tag{19}$$

$$E_{Linear}^{Ours} = T \times C \times fr \times f_{in} \times f_{out} \times E_{AC}. \tag{20}$$

### A.8 COMPARISON OF INFERENCE SCHEMES

Compared to the I-LIF inference scheme (count-based spike representation) shown in Figure 5, our model with optimal spike representation consistently requires fewer timesteps across all layers. This reduction directly translates into lower latency and energy consumption during inference, highlighting its practical advantages for efficient deployment.

### A.9 EVOLUTION OF MAXIMUM MEMBRANE POTENTIAL ACROSS TRAINING EPOCHS

As shown in Figure 6, the maximum membrane potential of LMS neurons at each layer rises rapidly in the early training epochs, then increases more slowly and ultimately converges to a stable range across all datasets. This behavior aligns with our design objective, where neurons are encouraged to learn a maximum membrane potential that is sufficient for information representation, and once an appropriate value is reached, the subsequent training primarily focuses on refining the network within this stable regime.

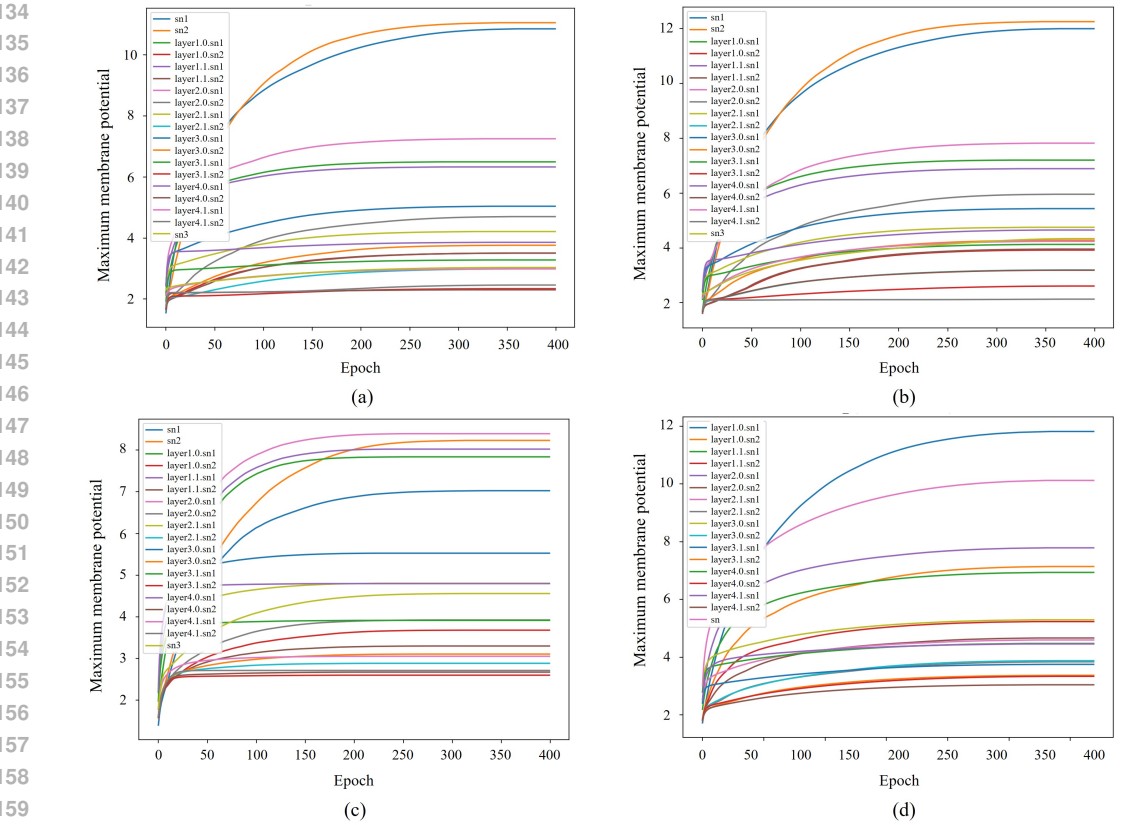

Figure 6: (a), (b), and (c) are the evolution of the maximum membrane potential across epochs on CIFAR10, CIFAR100, and CIFAR10-DVS using ResNet20, the corresponding timesteps are 1, 1, and 4, respectively. (d) is the evolution on ImageNet using ResNet18 (T=1).

## A.10 THEORETICAL ANALYSIS

We provide theoretical analysis of the proposed method from two perspectives: (1) following (Guo et al., 2024a; Wan et al., 2025), we show from an information-entropy viewpoint that Learning Maximum Spike can increase expressive capacity while reducing computational cost; and (2) we prove that the optimal spike representation is the optimality of the resulting encoding scheme.

**(1) Information-theoretic analysis of Learning Maximum Spike.** Given a discrete set $X$, its expressive capacity can be measured by the information entropy defined on $X$:

$$R(X) = \max \ H(X) = \max(-\sum_{x \in X} p_X(x) log p_X(x)), \tag{21}$$

where $p_X(x)$ denotes the probability of a sample $x$ from $X$. If there are $N$ possible values $\{x_1, x_2, \cdots, x_N\}$, and $p_X(x_1) = p_X(x_2) = \cdots = p_X(x_N)$, then $H(X)$ attains its maximum $logN$, i.e., $R(X) = logN$.

For a conventional binary spike feature map $F_B \in \{0,1\}^{C \times H \times W}$, the number of possible configurations is $2^{C \times H \times W}$. Thus, $R(F_B) = log2^{C \times H \times W}$. For our LMS, if the maximum integer spike in the current layer is $D$, then the corresponding integer spike feature map $F_D \in \{0, 1, \cdots, D\}^{C \times H \times W}$, leading to $R(F_D) = log(D+1)^{C \times H \times W}$. In our experiments, most layers satisfy $D \geq 4$ (See Table 9 in the appendix). Thus, LMS provides a substantially higher expressive capacity than conventional binary spike representations.

It is important to note that the effective expressive capacity of a layer is determined by the actual activation distribution $p_z$, rather than by the nominal upper bound $D$ alone. Intuitively, if we artificially fix a large maximum integer spike (e.g., D=16), the theoretical capacity upper bound

becomes strictly larger than that with D=4, i.e., $\log 17^{C \times H \times W} > \log 5^{C \times H \times W}$. However, even with this large theoretical bound, if after training almost all activations are concentrated in the sub-range $\{0, \cdots, 4\}$, then the layer effectively uses only 5 firing values, and the effective capacity of the feature map remains close to $\log 5^{C \times H \times W}$. In other words, those high integer spike that almost never occur do not provide a meaningful gain in expressive capacity, but they do increase the maximum required timesteps and introduce additional computational overhead.

The design goal of LMS is to make the learned maximum integer spike D adaptively match the effective dynamic range of the activation distribution $p_z$, so that each layer can learn an optimally sufficient expressive capacity while avoiding unnecessary computational overhead.

**(2) Optimality among all encodings.** We first establish that our optimal spike representation admits at least one feasible solution.

When $C = \lceil \log_2(D+1) \rceil$, we can choose standard binary weights:

$$w_j = 2^{j-1}, j = 1, \cdots, C. \tag{22}$$

For any integer $z \in \{0, \cdots, D\}$, there exists a unique set of binary coefficients $\{b_{j,z}\}$ such that

$$z = \sum_{j=1}^{C} 2^{j-1} b_{j,z}, b_{j,z} \in \{0, 1\}. \tag{23}$$

Thus, $\{w_j\}$ and $\{b_{j,z}\}$ together form a solution that satisfies the reconstruction constraint for all z. This shows that, when $C = \lceil \log_2(D+1) \rceil$, the optimal spike representation problem always admits at least one feasible solution.

Then, let $\mathcal{F}(C)$ denote the set of all encodings with length-C:

$$\mathcal{F}(C) = \left\{ (\{w_j\}, \{b_{j,z}\}) | \sum_{j=1}^{c} w_j b_{j,z} = z, \forall z, b_{j,z} \in \{0, 1\} \right\}. \tag{24}$$

On this set, we minimize

$$\mathcal{J}(\{w_j\}, \{b_{j,z}\}) = \sum_{z=0}^{D} p_z \sum_{j=1}^{C} b_{j,z}. \tag{25}$$

By definition, optimal spike representation computes a minimizer $(\{w_j^*\}, \{b_{j,z}^*\}) \in \mathcal{F}(\mathcal{C})$, so for any other feasible encoding $(\{w_j\}, \{b_{j,z}\}) \in \mathcal{F}(\mathcal{C})$, we have

$$\mathcal{J}(\{w_j^*\}, \{b_{j,z}^*\}) \leq \mathcal{J}(\{w_j\}, \{b_{j,z}\}). \tag{26}$$

In other words, among all encodings that exactly reconstruct the integer outputs with at most $C$ timesteps, our method is optimal in terms of expected spike count, thereby exhibiting the lowest energy consumption.

## A.11 IMPLEMENTATION ON NEUROMORPHIC CHIPS

Compared with standard SNNs, our method only needs one additional processing step: applying a weight to the feature map at each timestep after the convolution. This just relies on a global clock, which are already supported by mainstream neuromorphic chips (e.g., Tianjic (Pei et al., 2019), KA200 (Yang et al., 2024), and Loihi (Davies et al., 2018)).

## A.12 LIMITATION

The training of LMS involves two stages. The first stage is the standard SNN network training, which can take several days or even weeks, depending on the model size and hardware configuration. The second stage is the Optimal Spike Representation computation, which requires less than 10 seconds for all models. Compared to existing end-to-end deep learning frameworks, this approach is not dynamic and cannot adaptively determine the optimal model within a single training stage. Furthermore, there is substantial potential to further optimize the modeling of this IP problem and to design dedicated algorithms for its solution. This direction is highly promising for advancing discrete optimization in SNNs and promoting the development of more efficient and scalable SNNs. Finally, LMS introduces additional MAC operations, and at this stage we do not have a purely algorithmic way to completely eliminate them. We plan to address this issue in future work.

## A.13 THE USE OF LARGE LANGUAGE MODELS

We leveraged a large language model to polish the writing of our manuscript.

