# OpenReview forum: "LMS: Learnable Maximum Spike with Optimal Spike Representation for High-Performance and Efficient Spiking Neural Networks"
_ICLR.cc/2026/Conference — Submitted to ICLR 2026_

### Official Review · Reviewer_WVvo · 2025-10-20

**Soundness:** 3
**Presentation:** 3
**Contribution:** 3
**Rating:** 4
**Confidence:** 4

**Summary:**

This paper presents an improvement over Integer LIF (I-LIF). I-LIF reduces training cost by using integer spikes during training and binary spikes during inference; this work goes one step further and makes the maximum integer value trainable and layer-specific. The maximum is updated online in a sliding-average manner similar to the statistics used in Batch Normalization. After training, each spiking layer has a fixed, layer-wise integer spike maximum. To further cut inference cost, the authors propose an integer-programming-based algorithm that decomposes an integer spike into the shortest possible weighted sequence of binary spikes. Experiments on several datasets show lower energy, lower latency, and higher accuracy than previous methods.

**Strengths:**

1. The paper is clearly written, the motivation is sound, and the methodological details are well described.
2. State-of-the-art results are obtained on multiple datasets, confirming the effectiveness of the approach.

**Weaknesses:**

1. Compared with I-LIF inference, the proposed method appears to require a distinct weight for every time-step during inference. For long sequences, does this incur large memory overhead to store these per-step weights? Moreover, does this prevent the method from generalizing to variable-length sequences? In general, decomposing spikes along the time axis seems questionable—one could directly binarize ANN activations, yielding an “SNN” that is hardly different from the original ANN.
2. Source code is not provided in the supplementary material, which reduces reproducibility.

**Questions:**

1. Are the reported inference results obtained with the integer spikes or with the decomposed binary spikes?
2. The ImageNet accuracy is noticeably lower than that of recent SNN Transformer architectures. Why were those architectures not tested?

---

> ### Author Response · Authors · 2025-11-21
> **Response to Reviewer WVvo Part (1/2)**
>
> We sincerely appreciate the time you have dedicated to reviewing our manuscript and for providing such constructive feedback. We will address each of your concerns.
>
> > *W1: Compared with I-LIF inference, the proposed method appears to require a distinct weight for every time-step during inference. For long sequences, does this incur large memory overhead to store these per-step weights? Moreover, does this prevent the method from generalizing to variable-length sequences? In general, decomposing spikes along the time axis seems questionable—one could directly binarize ANN activations, yielding an “SNN” that is hardly different from the original ANN.*
>
> *Response to W1*: We thank the reviewer for the insightful comments and apologize for any fusion caused by our original description.
>
> (1)	Here we clarify the additional parameters / memory overhead compared with I-LIF. During inference, each training timestep is expanded into $C$ inference timesteps. This only introduces **$C$ parameters per layer**, which are shared across each training timestep. For long-sequence tasks, the situation is the same: if the sequence length is $Q$, inference runs for $Q \times C$ steps, but the number of additional parameters to store remains $C$ (See Table 9 in the appendix, our method typically use $C=3$) and **does not grow with $Q$**. This is a negligible memory overhead.
>
> (2)	For an input sequence of any length $Q$, we only need to store $C$ parameters, and the number of temporal weights does not depend on $Q$. Thus, our method can **generalize** to variable-length sequences.
>
> (3)	Finally, regarding the concern "decomposing spikes along the time axis seems questionable," we respectfully clarify that encoding firing strength within a time window by multiple discrete spikes is well grounded in neuroscience [1], and are explored in prior SNN works [2-4].
>
> As for the suggestion that "one could directly binarize ANN activations, yielding an ‘SNN’ that is hardly different from the original ANN". We respond from three aspects.
>
> (a)	The key distinction between ANNs and SNNs is that SNNs inherently model temporal dynamics [5].
>
> (b)	This idea is related to the well-established field of ANN-to-SNN conversion. However, ANN-to-SNN conversion approaches usually **cannot fully exploit the temporal dynamics** of SNNs during training [6-7] (e.g., membrane potential integration, threshold firing, and reset), and they often **require a large number of timesteps** at inference to approach the accuracy of the original ANN [8-9].
>
> (c)	Our method remains **biologically plausible**, fully leverages SNN **temporal dynamics**, and empirically achieves **higher accuracy with fewer timesteps, lower memory footprint, and shorter inference time.**
>
> ---
>
> > *W2: Source code is not provided in the supplementary material, which reduces reproducibility.*
>
> *Response to W2*: Thanks for the advice. We include the source code in the supplementary material to facilitate reproduction of our experimental results.

---

> > ### Author Response · Authors · 2025-11-21
> > **Response to Reviewer WVvo Part (2/2)**
> >
> > > *Q1: Are the reported inference results obtained with the integer spikes or with the decomposed binary spikes?*
> >
> > *Response to Q1*: Thanks for the question. Our reported inference results are obtained **using the decomposed binary spikes.**
> >
> > ---
> >
> > > *Q2: The ImageNet accuracy is noticeably lower than that of recent SNN Transformer architectures. Why were those architectures not tested?*
> >
> > Response to Q2: Thank you for your valuable suggestion. We additionally conduct ImageNet experiments with E-SpikeFormer [2] (a Transformer-based SNN architecture) to directly validate the effectiveness of LMS. The results are summarized in the table below:
> >
> > |Method|Param|Energy|T×C|Accuracy|
> > |---|---|---|---|---|
> > |E-SpikeFormer|5.1M|1.7mJ |1×4|75.3%|
> > | E-SpikeFormer+LMS (Ours)|5.1M|1.3mJ|1×3.19|76.6%|
> >
> > With 5.1M parameters, **our method achieves 76.6\% accuracy with 3.19 average inference timesteps, outperforming the vanilla E-SpikeFormer (75.3\% with 4 inference timesteps) by $+1.3\%$ accuracy. Moreover, our method requires only 1.3mJ energy consumption, achieving a 24\% reduction compared to the original model.** We incorporate these results into the Section 5.3 of main text, which further demonstrates that our method is effective across both CNN-based and Transformer-based SNN architectures.
> >
> > We hope our responses clarify the reviewer’s concerns and would be happy to address any further questions.
> >
> > ---
> >
> > [1] Fan L, et al. A multisynaptic spiking neuron for simultaneously encoding spatiotemporal dynamics. Nature Communications 2025.
> >
> > [2] Yao M, et al. Scaling spike-driven transformer with efficient spike firing approximation training. TPAMI 2025.
> >
> > [3] Lei Z, et al. Spike2former: Efficient spiking transformer for high-performance image segmentation. AAAI 2025.
> >
> > [4] Qiu X, et al. Quantized spike-driven transformer. ICLR 2025.
> >
> > [5] Roy K, Jaiswal A, Panda P. Towards spike-based machine intelligence with neuromorphic computing. Nature 2019.
> >
> > [6] Wan G, et al. Spik-NeRF: Spiking Neural Networks for Neural Radiance Fields. NeurIPS 2025.
> >
> > [7] Yu K, et al. TS-SNN: Temporal Shift Module for Spiking Neural Networks, ICML 2025.
> >
> > [8] Yu Q, et al. Constructing accurate and efficient deep spiking neural networks with double-threshold and augmented schemes. IEEE TNNLS 2021.
> >
> > [9] Liu F, et al. Spikeconverter: An efficient conversion framework zipping the gap between artificial neural networks and spiking neural networks. AAAI 2022.

---

> > > ### Comment · Reviewer_WVvo · 2025-11-21
> > >
> > > Thank you for your reply—most of my questions have been resolved. I still have two remaining questions. First, in Table 3, compared with previous works, this paper uses the same network structure but a larger number of time steps (T*C). Will this lead to higher energy consumption? Second, suppose I perform 8-bit quantization on the activation values of an ANN and decompose them into 8 binary pulses, then adopt standard binary encoding and decoding between layers—will this method outperform the method proposed in this paper in terms of performance?

---

> > > > ### Author Response · Authors · 2025-11-24
> > > > **Response to Reviewer WVvo**
> > > >
> > > > Thank you very much for your insightful feedback.
> > > > > *Q3: First, in Table 3, compared with previous works, this paper uses the same network structure but a larger number of time steps (T×C). Will this lead to higher energy consumption?*
> > > >
> > > > *Response to Q3*: Thank you for this valuable question. We compare LMS with LIF neurons (most methods adopt LIF neurons) under the same network architecture (ResNet20) on CIFAR10-DVS. The results are shown in the table below.
> > > >
> > > > |Model|T×C|Accuracy|Energy|
> > > > |---|---|---|---|
> > > > |LIF|10×1|79.50%|0.39mJ|
> > > > |LMS|4×2.79|82.70% (**+3.20%**)|0.18mJ (**-54%**)|
> > > > |LMS|10×3.10|83.83% (**+4.33%**)|0.47mJ (**+20%**)|
> > > >
> > > > With slightly larger average inference timesteps (4×2.79 vs. 10×1), LMS significantly reduces energy consumption (0.18 mJ vs. 0.39 mJ, -54%).
> > > >
> > > > With much larger timesteps (10×3.1 vs. 10×1), LMS’s energy consumption can exceed prior methods (0.47 mJ vs. 0.39 mJ, +20%).
> > > >
> > > > ---
> > > >
> > > > > *Q4: Second, suppose I perform 8-bit quantization on the activation values of an ANN and decompose them into 8 binary pulses, then adopt standard binary encoding and decoding between layers—will this method outperform the method proposed in this paper in terms of performance?*
> > > >
> > > > *Response to Q4*: Thank you for your question. We compare our method with an ANN using 8-bit activation quantization on CIFAR100. The results are reported in the table below (the quantized ANN’s energy is computed by adopting standard binary encoding and decoding between layers).
> > > >
> > > > |Model|Model|Accuracy|Energy|
> > > > |---|---|---|---|
> > > > |8-bit activation quantization|ResNet20|76.13%|0.16mJ|
> > > > |LMS|ResNet20|76.45% (**+0.32%**)|0.08mJ (**-50%**)|
> > > > |8-bit activation quantization|ResNet19|81.47%|1.23mJ|
> > > > |LMS|ResNet19|81.93% (**+0.46%**)|0.46mJ (**-63%**)|
> > > >
> > > > LMS achieves comparable performance while being substantially more energy-efficient than an ANN with 8-bit activation quantization.
> > > >
> > > > In addition, we further clarify the advantages of our method compared with activation quantization methods:
> > > >
> > > > 1.	The **dynamic**, learnable spike maximum of LMS makes it more **flexible** than activation quantization methods with fixed levels.
> > > >
> > > > 2. The inherent spike **sparsity** of SNNs enables our method to attain lower energy consumption compared to activation quantization methods.
> > > >
> > > > We hope our responses clarify your concerns and would be glad to address any further questions.

---

> > > > > ### Comment · Reviewer_WVvo · 2025-11-24
> > > > >
> > > > > Thanks for your reply. I have raised my rating.

---

> > > > > > ### Author Response · Authors · 2025-11-25
> > > > > >
> > > > > > Thanks very much for your reply and recognition. We are happy to see that your concerns have been addressed.

---

### Official Review · Reviewer_k7Bw · 2025-10-25

**Soundness:** 3
**Presentation:** 3
**Contribution:** 3
**Rating:** 6
**Confidence:** 5

**Summary:**

This paper proposes the Learnable Maximum Spike (LMS) neuron for high-performance, efficient Spiking Neural Networks (SNNs). Addressing binary spike-induced information loss, LMS emits integers during training, dynamically learns layer-specific maximum membrane potential (via membrane potential distribution), and uses a cosine-decayed balancing coefficient for stable training. It also formulates integer-to-binary spike representation as an integer programming problem to minimize inference energy. Experiments on CIFAR10/100, ImageNet, and CIFAR10-DVS show LMS outperforms SOTA methods, with lower inference memory (-7.16%), time (-12.09%), and energy (-61.9%).

**Strengths:**

First, it dynamically learns layer-specific spike maxima via membrane potential distribution, balancing SNN expressive capacity and computational efficiency.

Second, its cosine-decayed balancing coefficient stabilizes training by adapting to early/late-stage membrane potential changes.

Third, it minimizes inference energy via integer programming-based optimal spike representation, cutting energy by 61.9%.

**Weaknesses:**

First, it should supplement comparisons with more latest SOTA methods to better highlight LMS’s advantages in diverse scenarios.

Second, the brain diagram shown in Figure 1 is unnecessary as it doesn’t aid in explaining LMS’s core mechanisms.

**Questions:**

see Weaknesses

---

> ### Author Response · Authors · 2025-11-21
> **Response to Reviewer k7Bw**
>
> We are grateful to the reviewer for the detailed and insightful feedback. We respond to each concern below.
>
> > *W1: First, it should supplement comparisons with more latest SOTA methods to better highlight LMS’s advantages in diverse scenarios.*
>
> *Response to W1*: Thank you for this helpful comment. In the revised manuscript, **we add comparisons with more recent SOTA methods [1-4] in Table 1 and Table 2 to better highlight the advantages of LMS.** Our method still outperforms these SOTA methods. For example, on ImageNet with the ResNet18 architecture, **LMS surpasses ReverB-SNN [1] by 1.82% accuracy.**
>
> Moreover, to demonstrate the applicability of LMS in diverse scenarios, we also evaluate it on an object detection task by integrating LMS into SpikeYOLO [5] on COCO dataset. The experimental results are shown in the table below:
>
> |Method|Param|Energy|T×C|map@50|map@50:95|
> |---|---|---|---|---|---|
> |SpikeYOLO |23.1M|34.6mJ |1×4|62.3%|45.5%|
> | SpikeYOLO+LMS (Ours)|23.1M|24.7mJ|1×3.15|63.4%|46.7%|
>
> With 23.1M parameters, **LMS achieves 63.4\% mAP@50 and 46.7\% mAP@50:95 with 3.15 average inference timesteps, outperforming SpikeYOLO with 4 inference timesteps by +1.1\% mAP@50 and +1.2\% mAP@50:95. Moreover, our method requires only 24.7mJ energy consumption, compared to 34.6mJ for SpikeYOLO.** The corresponding results are included in the Section 5.3 of main text.
>
> ---
>
> > *W2: Second, the brain diagram shown in Figure 1 is unnecessary as it doesn’t aid in explaining LMS’s core mechanisms.*
>
> *Response to W2*:  Thank you for pointing out this issue. We remove the brain diagram from Figure 1 in the revised manuscript.
>
> We hope our responses clarify the reviewer’s concerns and would be happy to address any further questions.
>
> ---
>
> [1] Guo Y, et al. ReverB-SNN: Reversing Bit of the Weight and Activation for Spiking Neural Networks. ICML 2025.
>
> [2] Xue D, et al. MI-TRQR: Mutual Information-Based Temporal Redundancy Quantification and Reduction for Energy-Efficient Spiking Neural Networks. NeurIPS 2025.
>
> [3] Dong Y, Zhao D, Zeng Y. Temporal knowledge sharing enable spiking neural network learning from past and future. IEEE TAI 2024.
>
> [4] Yu K, et al. TS-SNN: Temporal Shift Module for Spiking Neural Networks, ICML 2025.
>
> [5] Luo X, et al. Integer-valued training and spike-driven inference spiking neural network for high-performance and energy-efficient object detection. ECCV 2024.

---

> > ### Comment · Reviewer_k7Bw · 2025-11-22
> >
> > Thank you for the authors' detailed responses—all my previous concerns have been fully addressed. The newly supplemented experiments effectively demonstrate the generalization ability of the proposed method, with notable advantages specifically in detection tasks. I am willing to raise my evaluation score to express my support for this work.

---

> > > ### Author Response · Authors · 2025-11-24
> > >
> > > Thanks very much for your reply and recognition. We are happy to see that your concerns have been addressed.

---

### Official Review · Reviewer_fKnC · 2025-10-30

**Soundness:** 4
**Presentation:** 4
**Contribution:** 4
**Rating:** 6
**Confidence:** 5

**Summary:**

This paper introduces a Learnable Maximum Spike (LMS) neuron. Based on each layer’s membrane-potential distribution, the network adaptively learns that layer’s maximum allowable integer spike. To balance rapid distribution tracking early in training with stable convergence later, a cosine-annealed balancing coefficient dynamically adjusts the update rate. At inference, integer outputs are converted into binary spike sequences by formulating an integer programming problem that incorporates firing-rate considerations to minimize energy. Across multiple datasets, LMS achieves higher or comparable accuracy with fewer inference steps, while also reducing memory usage, latency, and energy consumption.

**Strengths:**

1.The motivation is clear. Given that I-LIF is a strong baseline, revealing its limitations and proposing substantial improvements is important. Replacing I-LIF with the LMS neuron can improve accuracy while reducing energy.

2.The proposed per-layer scheme that learns the maximum integer spike a neuron may emit is conceptually simple and methodologically transparent. Its parsimony, together with the observed gains, makes the approach academically compelling.

3.Casting spike representation as an integer programming problem with an energy-minimization objective fills an efficiency-optimization gap that prior work has not fully addressed.

4.Extensive experiments validate the effectiveness of the method.

**Weaknesses:**

1.LMS introduces additional MACs, but the paper does not report the proportion of total energy consumption attributable. If that share is large, it would undermine the original motivation for SNNs.

2.The integer-programming solve is performed once after training, yet its solving time as a fraction of the total training time remains unclear.


3.The x-axis labels in Figure 4 are too small to read. Figures 1 and 3 appear partially redundant. Please justify the necessity of both or consider consolidating them.

**Questions:**

1.As dataset complexity increases, do neurons learn larger maximum integer spike?

2.In I-LIF, is $C$ set equal to $D$? The reported I-LIF inference appears not to use the optimal spike representation. This should be stated explicitly in the caption of Table 4 to avoid confusion.

---

> ### Author Response · Authors · 2025-11-21
> **Response to Reviewer fKnC**
>
> Thank you very much for your recognition of our work and your insightful review. We will address each of your questions.
>
> > *W1: LMS introduces additional MACs, but the paper does not report the proportion of total energy consumption attributable. If that share is large, it would undermine the original motivation for SNNs.*
>
> *Response to W1*: Thank you for your valuable suggestion. In the energy numbers reported for our method in Table 4, our method has a total inference energy of **0.08 mJ**. The additional MAC operations introduced by our method consume about **0.004mJ**, which is only **5%** of the total energy. This small fraction does not undermine the energy advantage of our method compared with prior approaches.
>
> ---
>
> > *W2: The integer-programming solve is performed once after training, yet its solving time as a fraction of the total training time remains unclear.*
>
> *Response to W2*: Thank you for raising this point. The IP solving time on different models is summarized below:
>
> |Dataset|Model|GPU|Total training time|IP solving time|
> |---|---|---|---|---|
> |CIFAR100|ResNet20|Single RTX 4090|17580s|12.12s|
> |ImageNet|ResNet34|Single RTX 4090|326080s|19.01s|
>
> For CIFAR100 with ResNet20, the training stage requires approximately **17,580s**, whereas the IP solver completes in **12.12s**, accounting for only **0.069%** of the total training time. For ImageNet with ResNet34, training takes about **326,080s**, and the IP solve finishes in **19.01s**, corresponding to merely **0.0058%** of the overall training time. These results indicate that the IP solver overhead is negligible. We provide these details in Appendix A3.1 (line 881-886).
>
> ---
>
> > *W3: The x-axis labels in Figure 4 are too small to read. Figures 1 and 3 appear partially redundant. Please justify the necessity of both or consider consolidating them.*
>
> *Response to W3*: Thank you for your valuable suggestions and guidance. For Fig. 4, we increase the font size of the x-axis labels to make them easier to read (the revised figure is updated in the manuscript).
>
> What’s more, we consider it necessary to keep both Fig. 1(c) and Fig. 3. **Fig. 1(c) illustrates the LMS neuron’s behaviors across layers and helps readers better understand how our method works from a layer-wise perspective, while Fig. 3 compares the LMS neuron with other neuron models and highlights the advantages of our approach.** Taken together, they serve different purposes, so it is necessary to keep both figures.
>
> ---
>
> > *Q1: As dataset complexity increases, do neurons learn larger maximum integer spike?*
>
> *Response to Q1*: Thanks for the question. **As the dataset becomes more complex, LMS indeed tends to learn slightly larger maximum integer spike values, and the corresponding average number of inference timesteps increases mildly.** This trend can be clearly observed in our experiments. For example:
>
> On CIFAR10, using ResNet20 trained with $T=1$, LMS yields an average of about 2.73 inference timesteps;
>
> On CIFAR100, with the same configuration (ResNet20, $T=1$), the average inference timesteps increase to about 2.95;
>
> On the more complex ImageNet dataset, a similar architecture (ResNet18) with $T=1$ has an average of about 3.00 inference timesteps.
>
> These results show that as dataset complexity increases from CIFAR-10 $\rightarrow$ CIFAR-100 $\rightarrow$ ImageNet, the learned maximum integer spikes and the corresponding average inference timesteps both increase slightly.
>
> **It is worth noting that even on the most complex dataset (ImageNet), our average number of inference timesteps remains lower than that of competing methods**, so the energy-efficiency and inference-efficiency advantages of LMS are still preserved.
>
> ---
>
> > *Q2: In I-LIF, is C set equal to D? The reported I-LIF inference appears not to use the optimal spike representation. This should be stated explicitly in the caption of Table 4 to avoid confusion.*
>
> *Response to Q2*: Thank you for pointing out this issue. In I-LIF, C is set equal to D. We explicitly clarify this in the caption of Table 4 in the revised manuscript to avoid confusion. Thanks again for pointing this out.

---

> > ### Comment · Reviewer_fKnC · 2025-11-26
> > **Response to authors**
> >
> > All my prior concerns have been fully and satisfactorily addressed. I am pleased to revise my evaluation score upward.

---

> > > ### Author Response · Authors · 2025-11-26
> > >
> > > Thanks very much for your reply and recognition. We are happy to see that your concerns have been addressed.

---

### Official Review · Reviewer_Fo7Y · 2025-11-01

**Soundness:** 3
**Presentation:** 3
**Contribution:** 2
**Rating:** 2
**Confidence:** 5

**Summary:**

This paper proposes the Learnable Maximum Spike (LMS) neuron, which is based on integer LIF (ILIF) neurons, for directly trained SNNs. To optimize the existing ILIF, the authors propose a method that optimizes ILIF based on the neuron's expression range. During training, the ILIF expression range is determined by the maximum value at each layer. Furthermore, for further optimization, they perform optimization using IP to minimize spike firing while reducing the inference time step. The proposed method has been validated on several image classification tasks.

**Strengths:**

1. Optimization for ILIF neurons. The optimization method proposed in this paper will be helpful for efficient application of ILIF.
2. Competitive empirical results at low time steps. The paper reports strong accuracy–efficiency trade-offs: e.g., ImageNet ResNet34 71.36% with T×C = 1×3.33.

**Weaknesses:**

Insufficient analysis of the optimal spike representation
- What is the overhead of the IP solver?
- As shown in Table 6, there are weights that are non-powers-of-two (3, 6). In this case, a MAC operation is required (Equation 11). Is there a way to eliminate the MAC operation? Is this operation included in energy analysis?
- What is the theoretical analysis for the proposed method?
- What is the decoding method for weighted spikes?

Lack of validation
- Lack of validation for applicability to various model architectures (e.g., Transformer)
- Lack of validation for applicability to tasks other than image classification

Lack of direct discussion of neuromorphic hardware
The paper states that the goal of SNNs is energy efficiency on neuromorphic hardware, but there is insufficient discussion on whether the proposed method is feasible on neuromorphic hardware.

**Questions:**

Please refer to Weaknesses section.

---

> ### Author Response · Authors · 2025-11-21
> **Response to Reviewer Fo7Y Part (1/3)**
>
> We are grateful to the reviewers for their detailed and insightful feedback. We will answer your questions point by point below.
>
> > *W1: What is the overhead of the IP solver?*
>
> *Response to W1*: Thank you for this question. The IP is performed only once per model after training is complete. Its solving time on different models is summarized below:
>
> |Dataset|Model|GPU|Total training time|IP solving time|
> |---|---|---|---|---|
> |CIFAR100|ResNet20|Single RTX 4090|17580s|12.12s|
> |ImageNet|ResNet34|Single RTX 4090|326080s|19.01s|
>
> For CIFAR100 with ResNet20, the training stage requires approximately **17,580s**, whereas the IP solver completes in **12.12s**, accounting for only **0.069%** of the total training time. For ImageNet with ResNet34, training takes about **326,080s**, and the IP solve finishes in **19.01s**, corresponding to merely **0.0058%** of the overall training time. These results indicate that the IP solver overhead is negligible. We provide these details in Appendix A3.1 (line 881-886).
>
> This is expected from our formulation. For a given layer $l$, the IP involves a number of decision variables and linear constraints proportional to $C_l(D_l+1)$. Hence, the overall size of all IPs across the network depends only on $\{C_l, D_l\}$ and the number of layers $L$. In all our experiments, both $C_l$ and $D_l$ are small integers, so the additional computation introduced by the IP step is very modest compared to training the model.
>
> ---
>
> > *W2: As shown in Table 6, there are weights that are non-powers-of-two (3, 6). In this case, a MAC operation is required (Equation 11). Is there a way to eliminate the MAC operation? Is this operation included in energy analysis?*
>
> *Response to W2*: Thanks for your question.
>
> (a)	We do not have a purely algorithmic way to completely eliminate the MAC operation. We discuss this as a limitation in Appendix A.12 and regard it as a direction for future work.
>
> (b)	The overhead of this MAC operation is fully **included** in our method’s energy analysis. The **0.08 mJ** reported for our method in Table 4 includes **0.004 mJ** from the additional MAC operations (**5%**). We state in the original manuscript that our method introduces a small number of additional MAC operation (line 430).

---

> ### Author Response · Authors · 2025-11-21
> **Response to Reviewer Fo7Y Part (2/3)**
>
> > *W3: What is the theoretical analysis for the proposed method?*
>
> *Response to W3*: Thank you for this helpful comment. We provide theoretical analysis of the proposed method from two perspectives: (1) following [1, 2], we show from an **information-entropy viewpoint** that Learning Maximum Spike can increase expressive capacity while reducing computational cost; and (2) we prove that the optimal spike representation is **the optimality of the resulting encoding scheme.**
>
> (1)	**Information-theoretic analysis of Learning Maximum Spike.**
> Given a discrete set $X$, its expressive capacity can be measured by the information entropy defined on $X$:
>
> $R(X)=\max \ H(X) = \max (-\sum_{x \in X}p_X(x)logp_X(x)),$
>
> where $p_X(x)$ denotes the probability of a sample $x$ from $X$. If there are $N$ possible values $\{x_1, x_2,\cdots, x_N\}$, and $p_X(x_1)= p_X(x_2)=\cdots= p_X(x_N)$, then $H(X)$ attains its maximum $logN$, i.e., $R(X) = logN$.}
>
> For a conventional binary spike feature map $F_B \in \\{0, 1\\}^{C \times H \times W}$, the number of possible configurations is $2^{C \times H \times W}$. Thus, $R(F_B) = log 2^{C \times H \times W}$. For our LMS, if the maximum integer spike in the current layer is $D$, then the corresponding integer spike feature map $F_D \in \\{0, 1,\cdots,D\\}^{C \times H \times W}$, leading to $R(F_D) = log (D+1)^{C \times H \times W}$. In our experiments, most layers satisfy $D \ge 4$ (See \Cref{tab:layer-params} in the appendix). Thus, LMS provides a substantially higher expressive capacity than conventional binary spike representations.
>
> It is important to note that the effective expressive capacity of a layer is determined by the actual activation distribution $p_z$, rather than by the nominal upper bound $D$ alone. Intuitively, if we artificially fix a large maximum integer spike (e.g., $D=16$), the theoretical capacity upper bound becomes strictly larger than that with $D=4$, i.e., $\log17^{C\times H\times W} > \log5^{C\times H\times W}$. However, even with this large theoretical bound, if after training almost all activations are concentrated in the sub-range $\\{0,\cdots,4\\}$, then the layer effectively uses only 5 firing values, and the effective capacity of the feature map remains close to $\log 5^{C \times H \times W}$. In other words, those high integer spike that almost never occur do not provide a meaningful gain in expressive capacity, but they do increase the maximum required timesteps and introduce additional computational overhead.
>
> The design goal of LMS is to make the learned maximum integer spike $D$ adaptively match the effective dynamic range of the activation distribution $p_z$, so that each layer can learn an optimally sufficient expressive capacity while avoiding unnecessary computational overhead.
>
> (2) **Optimality among all encodings.**
> We first establish that our optimal spike representation admits at least one feasible solution.
>
> When $C = \lceil \log_2(D + 1) \rceil$, we can choose standard binary weights:
>
>  $w_j=2^{j-1},j=1,\cdots,C.$
>
> For any integer $z \in \\{0,\cdots,D\\}$, there exists a unique set of binary coefficients $\\{b_{j,z}\\}$ such that
>
> $z=\sum_{j=1}^C 2^{j-1} b_{j,z},b_{j,z} \in \\{0,1\\}.$
>
> Thus, $\\{w_j\\}$ and $\\{b_{j,z}\\}$ together form a solution that satisfies the reconstruction constraint for all $z$. This shows that, when $C = \lceil \log_2(D + 1) \rceil$, the optimal spike representation problem always admits at least one feasible solution.
>
> Then, let $\mathcal{F}(C)$ denote the set of all encodings with length-C:
>
> $\mathcal{F}(C)=\left\\{(\\{w_j\\},\\{b_{j,z}\\})|\sum_{j=1}^cw_jb_{j,z}=z,\forall z,b_{j,z}\in\\{0,1\\}\right\\}.$
>
> On this set, we minimize
>
> $\mathcal{J}(\{w_j\},\{b_{j,z}\})=\sum_{z=0}^Dp_z\sum_{j=1}^Cb_{j,z}.$
>
> By definition, optimal spike representation computes a minimizer $(\\{w_j^\*\\},\\{b_{j,z}^\*\\})\in\mathcal{F}(\mathcal{C})$, so for any other feasible encoding $(\\{w_j\\},\\{b_{j,z}\\})\in\mathcal{F}(\mathcal{C})$, we have
>
> $\mathcal{J}(\\{w_j^\*\\},\\{b_{j,z}^\*\\})\leq\mathcal{J}(\\{w_j\\},\\{b_{j,z}\\}).$
>
> In other words, among all encodings that exactly reconstruct the integer outputs with at most $C$ timesteps, our method is optimal in terms of expected spike count, thereby exhibiting the lowest energy consumption.
>
> We summarize these theoretical results in the revised version (Appendix A.10) to better clarify why Learning Maximum Spike and the proposed Optimal Spike Representation jointly provide stronger representational capacity while being more energy-efficient.

---

> > ### Author Response · Authors · 2025-11-21
> > **Response to Reviewer Fo7Y Part (3/3)**
> >
> > > *W4: What is the decoding method for weighted spikes?*
> >
> > *Response to W4*:  Thanks for your question. As shown in Eq. (11), once the integer spikes have been encoded into 0/1 and processed by the convolution operation, we obtain one feature map at each expanded timestep, and the decoded output is computed by a weighted sum over these feature maps.
> >
> > More concretely, each original training timestep is expanded into $C$ extended-timesteps during inference. For the $t$-th original timestep, after 0/1 encoding and convolution, we obtain $C$ feature maps $\\{F_{t,1}, \cdots, F_{t,C}\\}$. We then multiply each expanded-timestep feature map $F_{t,j}$ by its corresponding weight $w_j$ and sum them:
> >
> > $\tilde{F_t} = \sum_{j=1}^{C} w_j F_{t,j}.$
> >
> > In this way, we decode one output feature $\tilde{F}_t $ for each original timestep $t$.
> >
> > ---
> >
> > > *W5: Lack of validation for applicability to various model architectures (e.g., Transformer)*
> >
> > *Response to W5*: Thank you for pointing out this issue. We validate the applicability of LMS with E-SpikeFormer [3] (a Transformer-based architecture) on ImageNet dataset. The results are shown in the table below:
> >
> > |Method|Param|Energy|T×C|Accuracy|
> > |---|---|---|---|---|
> > |E-SpikeFormer|5.1M|1.7mJ |1×4|75.3%|
> > | E-SpikeFormer+LMS (Ours)|5.1M|1.3mJ|1×3.19|76.6%|
> >
> > With 5.1M parameters, **our method achieves 76.6\% accuracy with 3.19 average inference timesteps, outperforming the vanilla E-SpikeFormer (75.3\% with 4 inference timesteps) by $+1.3\%$ accuracy. Moreover, our method requires only 1.3mJ energy consumption, achieving a 24\% reduction compared to the original model.** We incorporate these results into the Section 5.3 of main text, which further demonstrates that LMS is broadly applicable across different network architectures, including Transformer-based SNNs.
> >
> > ---
> >
> > > *W6: Lack of validation for applicability to tasks other than image classification*
> >
> > *Response to W6*: Thanks for your valuable suggestion. We integrate LMS into SpikeYOLO [4] for object detection on COCO dataset. The experimental results are shown in the table below:
> >
> > |Method|Param|Energy|T×C|map@50|map@50:95|
> > |---|---|---|---|---|---|
> > |SpikeYOLO |23.1M|34.6mJ |1×4|62.3%|45.5%|
> > | SpikeYOLO+LMS (Ours)|23.1M|24.7mJ|1×3.15|63.4%|46.7%|
> >
> > With 23.1M parameters, **LMS achieves 63.4\% mAP@50 and 46.7\% mAP@50:95 with 3.15 average inference timesteps, outperforming SpikeYOLO with 4 inference timesteps by +1.1\% mAP@50 and +1.2\% mAP@50:95. Moreover, our method requires only 24.7mJ energy consumption, compared to 34.6mJ for SpikeYOLO.** These results are included to the Section 5.3 of main text, further demonstrating that LMS is also effective on non-classification tasks such as object detection.
> >
> > ---
> >
> > > *W7: Lack of direct discussion of neuromorphic hardware. The paper states that the goal of SNNs is energy efficiency on neuromorphic hardware, but there is insufficient discussion on whether the proposed method is feasible on neuromorphic hardware.*
> >
> > *Response to W7*: Thanks for your advice. Compared with existing SNNs, our method only needs one additional processing step: applying a weight to the feature map at each inference timestep after the convolution. This just relies on a global clock, which are already **supported** by mainstream neuromorphic chips (e.g., Tianjic [5], KA200 [6], and Loihi [7]). We discuss the feasibility in the Appendix A.11.
> >
> > We hope our responses clarify the reviewer’s concerns and would be happy to address any further questions.
> >
> > ---
> >
> > [1] Guo Y, et al. Ternary spike: Learning ternary spikes for spiking neural networks. AAAI 2024.
> >
> > [2] Wan G, et al. Spik-NeRF: Spiking Neural Networks for Neural Radiance Fields. NeurIPS 2025.
> >
> > [3] Yao M, et al. Scaling spike-driven transformer with efficient spike firing approximation training. TPAMI 2025.
> >
> > [4] Luo X, et al. Integer-valued training and spike-driven inference spiking neural network for high-performance and energy-efficient object detection. ECCV 2024.
> >
> > [5] Pei J, et al. Towards artificial general intelligence with hybrid Tianjic chip architecture. Nature 2019.
> >
> > [6] Yang Z, et al. A vision chip with complementary pathways for open-world sensing. Nature 2024.
> >
> > [7] Davies M, et al. Loihi: A neuromorphic manycore processor with on-chip learning. Ieee Micro, 2018.

---

> > > ### Author Response · Authors · 2025-11-28
> > >
> > > Thank you for your thoughtful comments and for the time you have invested in our work. We have posted our response and hope the above clarifications and the additional experiments sufficiently addressed your concerns. If any questions remain unresolved, please let us know. We would be grateful to clarify further and are happy to discuss specific points.

---

### Author Response · Authors · 2025-12-03
**Summary of Contribution and Rebuttal**

Dear ACs,

Thank you sincerely for taking the time and effort to handle our submission with such professionalism. We also greatly appreciate all reviewers' thoughtful and constructive feedback, which is invaluable in helping us improve the paper.

We are encouraged that the reviewers recognized the strengths of our work.

-	Reviewer fKnC notes that “the motivation is clear”, “fills an efficiency-optimization gap”, and “the approach academically compelling”.

-	Reviewer WVvo remarks that “the paper is clearly written, the motivation is sound, and the methodological details are well described”.

-	Reviewer k7Bw emphasizes that our method balances “SNN expressive capacity and computational efficiency”.

-	Reviewer Fo7Y highlights that our method achieves “strong accuracy-efficiency trade-offs.”

At the same time, we take the reviewers' suggestions seriously and further refine our work.

Below, we briefly summarize our main contributions and the improvements made in our rebuttal:

-	We make each layer adaptively learn its **own maximum spike** based on its membrane potential distribution, and introduce a cosine-decay balancing coefficient to stabilize training.

-	We formulate the optimal spike representation as an **integer programming problem** that minimizes energy consumption during inference.

-	Our method achieves simultaneous gains in **accuracy** and **energy efficiency** across a variety of tasks.

In addition, we carefully address all reviewers' concerns, including integer-programming solution time, further energy-consumption analysis, and new scalability experiments. After our responses, Reviewers fKnC, WVvo, and k7Bw **raise their scores to 8, 6, and 8 on Nov 25, Nov 24, and Nov 21 (AoE)**, respectively. Reviewer Fo7Y’s comments mainly focus on clarifications and additional analyses, which we also address one by one in the rebuttal.

We believe this new perspective opens a promising direction for understanding and addressing information expression in SNNs, while simultaneously improving performance and reducing energy consumption.

In light of these contributions, we sincerely hope you will consider supporting our submission.

Best regards,

Authors

---

### Meta-Review · Area_Chair_sAZA · 2026-01-06

**Summary:**

This paper proposes the Learnable Maximum Spike (LMS) neuron, an extension of integer LIF (I-LIF) neurons, where each layer adaptively learns its maximum integer spike based on membrane potential statistics, combined with a cosine-decayed update rule. At inference, the authors further formulate integer-to-binary spike decomposition as an integer programming (IP) problem to minimize energy consumption.

During the review process, reviewers generally agreed that the paper is clearly written, technically sound, and supported by extensive experiments. The rebuttal was thorough and responsive: the authors added additional theoretical discussion, overhead analysis of the IP solver, new experiments on Transformer-based architectures and object detection tasks, energy breakdowns, and clarifications on neuromorphic feasibility. As a result, several reviewers raised their scores and acknowledged that most technical questions were addressed.

However, despite the strong rebuttal and improved empirical coverage, the discussion converged on a more fundamental concern: the incremental nature of the contribution and its conceptual positioning within the broader SNN literature. While LMS improves upon I-LIF and related integer-spike formulations, reviewers expressed persistent doubts about whether the proposed method constitutes a sufficiently novel or foundational advance, as opposed to a careful engineering refinement of existing ideas. These concerns remain even after the rebuttal and ultimately inform the recommendation. Given the competitiveness of the venue, the work does not clearly meet the bar for acceptance at this time.

**Reviewer Concerns:**

Concerns addressed by the rebuttal:
- Implementation and overhead questions (Fo7Y, fKnC):
The authors clearly quantified the IP solver cost, showing it is negligible relative to training time, and provided detailed energy breakdowns, including the additional MAC operations.
- Scope of experimental validation (Fo7Y, k7Bw, WVvo):
New experiments on Transformer-based SNNs (E-SpikeFormer), object detection (SpikeYOLO), and additional SOTA comparisons substantially strengthened empirical coverage beyond image classification.
- Clarity and presentation issues (fKnC, k7Bw, WVvo):
Figure clarity, redundancy, architectural details, decoding procedures, and reproducibility concerns were adequately addressed, and reviewers acknowledged these improvements.

Concerns that remain outstanding:
- Limited conceptual novelty beyond existing integer-spike SNN frameworks:
The core ideas—integer spikes, adaptive range control, and spike decomposition for efficiency—are all natural extensions of prior I-LIF and integer-valued SNN work. While LMS integrates these components effectively, the paper does not clearly establish a new modeling paradigm or theoretical insight that fundamentally changes how spike-based computation is understood or designed.
- Blurring of boundaries between SNNs and quantized ANNs:
Despite the authors’ clarifications, the method’s reliance on integer activations, weighted binary decomposition, and post-training optimization raises unresolved questions about whether LMS meaningfully advances spike-based computation, or whether it primarily recasts quantized ANN techniques in SNN form. This concern was raised by multiple reviewers and remains only partially resolved.
- Energy-efficiency claims remain conditional:
While energy reductions are demonstrated, results show that under certain timestep configurations LMS can consume more energy than prior methods. Moreover, neuromorphic feasibility is discussed at a high level rather than demonstrated, leaving practical deployment advantages somewhat speculative.

**Reviewer Scores:**

Reviewer Fo7Y: Maintains a reject-level assessment despite clarifications, citing insufficient depth of analysis and conceptual concerns.

Reviewers fKnC, k7Bw, WVvo: Raised scores after rebuttal, but explicitly stated they would not mind if the paper were rejected, reflecting lingering reservations about contribution strength rather than correctness.

---

### Decision · Program_Chairs · 2026-01-26

Reject